# The Principal-Agent Theoretical Ramifications on Digital Transformation of Ports in Emerging Economies

Benjamin Mosses Sakita *, Berit Irene Helgheim and Svein Bråthen

Faculty of Logistics, Molde University College, Specialized University in Logistics, 6402 Molde, Norway; berit.i.helgheim@himolde.no (B.I.H.); svein.brathen@himolde.no (S.B.)
* Correspondence: benjamin.m.sakita@himolde.no

**Abstract:** *Background*: Scholarly literature indicates a slow pace at which maritime ports fully embrace digital transformation (DT). The reasons to this are largely anecdotal and lack solid empirical grounding. This inhibits an overall understanding of DT's tenets and the development of evidence-based policies and targeted actions. *Methods*: This study deployed a qualitative case study strategy to unpack the challenges of undertaking DT through the lens of principal-agent theory (PAT). *Results*: Analysis of data collected through 13 semi-structured interviews from a port's value chain stakeholders revealed five thematic challenges that contradict successful implementation of DT. These included interagency constraints and system ownership tussles; system sabotage and prevalent corruption; prevalent human agency in port operations; cultural constraints; and political influence on port governance. *Conclusions*: To address these challenges, the study proposes a four-stage empirically grounded DT strategy framework that guides both practitioners and policymakers through DT endeavors. The framework includes: (1) the port's value chain mapping, (2) stakeholder engagement, (3) resource mobilization, and (4) effective monitoring. For scholars, we provide an avenue for testing statistical significance of association and causality among the identified challenges.

**Keywords:** digital innovation diffusion; digital transformation strategy; stakeholder engagement; theory elaboration





## 1. Introduction

The burgeoning globalized markets, with a corresponding strong growth rate in international shipping trade at 3.2 percent, and the COVID-19 pandemic have made seaports and their related ecosystems ever-important logistics nodes in the global supply chains [1]. As an integral part of global supply chains that are increasingly becoming digitally transformed, most ports hitherto largely rely on manual processes and person-to-person systems that hinder their efficacy—through congestion and longer dwell times [2,3]. The extant literature informs that ports are traditionally slow in embracing digital transformation (DT) despite being critical linchpins in global economies and countries' economic hubs and gateways [4–10]. As global freight transport soars and modern ships transport more cargo, ports are under pressure to adjust both their physical and digital infrastructure and keep abreast. Likewise, populations in the developing world bulge yearly. For instance, in sub-Saharan Africa and eastern and southern Africa, the current growth rates are 2.5 percent and 2.1 percent, respectively [11]. This has accelerated the need for imported goods transiting third-world countries' frontiers where ports become in the spotlight. Meanwhile, Global South countries are typical sources of raw materials for the Western world, with about 762.4 million tons of goods loaded in 2021 [1]. In both scenarios, seaports in these economies become critical facilitators of international trade and a greater subject for efficiency improvements. The extant literature acknowledges notable differences in DT among ports in developing economies against their counterparts in developed economies [12]. This complicates the integration of global supply chains and stifles smooth

cargo and information flows [13]. For instance, a recent World Bank report on the global container ports performance index (CPPI) underscores the need for Africa's ports to improve operational efficiency. This will enable them to fully participate in international trade and commerce and facilitate greater food security on the continent [14].

DT presents seaports with new opportunities to manage their operations more efficiently and sustainably [15,16]. However, integrating port ecosystems with digital technologies has been slow. For instance, the extant literature associates such slowness with challenges such as the following: the existence of many siloed systems in the port ecosystem [17]; stakeholders' non-interoperable legacy systems [18]; missing standards regarding data sharing, transferring, and communication [5]; culture; investment cost; digital literacy; and lack of vision [3,9,19]. Unfortunately, these factors have largely been anecdotal and lack a rigorous empirical basis [20].

Specifically, there is a paucity of research studies on port DT endeavors in developing countries. For instance, most studies on DT hardly discourse on contextual factors such as politics and corruption. This may be influenced by their framework of reference (i.e., the developed world), where the preceding factors are less likely to be imminent challenges. Consequently, the models and theories that are developed only reflect applicability in developed contexts. Nevertheless, the existing literature generalizes relevant factors based on seaports in developed economies (i.e., [4,21,22]). These contextual factors may exhibit subtleties at technological levels and social and political ramifications compared to those existing in developing economies [12]. The lack of discourse on DT initiatives in developing economies precludes a rounded understanding of the peculiarities of DT under different contexts. This, therefore, warrants research that focuses on developing economies.

*Motivation*

In the rapidly evolving global trade landscape, the DT of ports stands as a pivotal frontier that may enable seamless integrations of key stakeholders in freight movements and handling. Meanwhile, the rate at which technological solutions are developed outsmarts the pace at which maritime organizations acquire and exploit them. This has caused many researchers to argue that the maritime industry and ports, in particular, have been slow in acquiring digital transformation. Consequently, over the past two and a half decades, researchers have ardently engaged in the exploration of the DT phenomenon in an attempt to unlock its tenets. This interest is evident not only in maritime-related literature but also across industries and industrial sectors. Thus, DT is a ubiquitous concept whose label has not yet found its home in many organizations, as not so much is known about it as of now. Some researchers have unveiled its tenets in terms of exploring the driving force behind its implementation [9,23], while others have explored the challenges that organizations face when implementing it [24–26]. Meanwhile, studies that investigate the enablers and success factors of DT in a port context are not uncommon either [23,27,28]. The common feature among these studies, however, is the presentation of anecdotal accounts of tenets that plausibly define DT. Up until mid-2023, only about 35 empirical literature on ports' DT had been available [23]. The dissection of such empirical literature reveals a convincingly limited exploration of the DT phenomenon as the majority of studies use mathematical modeling techniques such as AHP and FsQCA [12,29] and other methods such as business observation tools (BOT) and SWOT [30,31] to synthesize experts opinions on various aspects of undertaking successful DT in port settings. Very little evidence in the extant literature is a product of more robust methodological approaches that delve straight into user organizations of DT in attempts to explore the phenomenon thoroughly [3,4,21]. For instance, Sakita et al. [23] revealed that out of 35 empirical studies on ports DT, only 28.6 percent had used a qualitative case study approach. The emphasis on qualitative case studies cannot be stressed enough as the DT phenomenon is quite nascent, and its causal mechanism must be meticulously teased out. This can also be evidenced by a very low percentage of survey studies (i.e., 20%) indicating that DT as a concept is still nascent and its tenets have not yet been fully untangled.

It is arguably true that a novel phenomenon such as DT can only be comprehended better when it is explored from the perspectives of users or potential users. However, evidence that indicates this in the extant literature is still very limited. Consequently, anecdotal research, which largely uses unsubstantiated concepts to explain DT, provides broadly superficial policy and practical recommendations.

Against this background, this research addresses two objectives. Firstly, it responds to a call for more empirical research on DT in the maritime industry in general and ports in particular, as recommended by [9,20,23], by conducting an empirical inquiry into the various factors that exert influence on DT in emerging economies. Secondly, we frame maritime port's DT as principal-agent relationships between four stakeholder groups: national government, LPCOs, port authority, and customs authority. We use principal-agent theory (PAT) in accordance with Eisenhardt [32] to uncover the intricacies of interdependencies among maritime port stakeholders that challenge the successful implementation of DT initiatives. Therefore, this study's contributions are manifold: (1) conducting an empirical case study on ports in emerging economies provides a nuanced understanding of challenges and complexities involved in undertaking successful DT, thereby helping to develop more robust policy and practical recommendations; (2) we offer a four-staged empirically grounded framework of DT strategy for ports, which provides policy makers and port practitioners with tangible recommendations.

The reminder of this paper is structured as follows: Section 2 gives a literature review and sets the stage for the development of propositions through the lens of PAT. Section 3 elaborates on the methodology and strategies deployed to address the research question. Section 4 narrates the results in line with PAT's theoretical ramifications and proposes an emerging framework that details the challenges of undertaking successful DT in ports in emerging economy contexts. Section 5 places the findings in perspective and scrutinizes them against the backdrop of the propositions. Finally, Section 6 provides the conclusions, implications, recommendations, and areas for further research.

## 2. Literature and Theoretical Frame

### 2.1. DT and the Maritime Ports Sector

We define ports' DT as a continuous process through which port organizations either revamp or radically transform the means of value creation and transfer by exploiting novel technologies. DT is driven by such technologies as artificial intelligence, social media, cloud and edge computing, robotics, block chain, internet of things, 3D printing, and big data [20]. In practice, DT loosely connotes the exploitation of such technologies as radio frequency identification (RFID), sensors, and electronic single window systems (eSWS). Others include automation, port community systems, and electronic data interchange (EDI) [2,24,33]. The different interpretations of what DT connotes arguably translate into nuanced implementations of digital endeavors and, therefore, digital maturity levels across ports around the world [13].

The latest attempts to address the call to conduct more empirical research [9] have been made by Raza et al. [3], who performed an empirical investigation on DT with a particular focus on liner shipping using a qualitative case study design. Meanwhile, Gómez Díaz et al. [30] investigated the digital governance approach in the Spanish port system using qualitative business observation technique (BOT). Likewise, Hsu et al. [27] analyzed critical factors that influence smart port service quality using analytical hierarchy process (AHP) and decision-making trial and evaluation laboratory (DEMATEL) techniques. Similarly, Seo et al. [34] investigated digitalization strategies for container supply chains using AHP, fuzzy analytical hierarchy process (FAHP), and technique for order of preference by similarity to ideal solution (TOPSIS). Chowdhury et al. [29] investigated the barriers to implementing smart ports using the interpretive structural model (ISM) technique. The modeling approaches that dominate these empirical studies, while useful, hardly capture the contextual realities of actual actors in the maritime business.

*2.2. Related Literature on the Challenges of Undertaking Successful DT*

The literature that investigates the challenges of undertaking DT in seaports exists in leaps and bounds. Specific to technologies that constitute DT, various authors have made contributions in terms of literature reviews or empirical research. For instance, Gekara and Nguyen [25] investigated the challenges of implementing a container terminal operating system through a technology–organization–environment framework; the authors revealed that the lack of technological and data standards inhibited interoperability of newly installed CTOS, unstable broadband connectivity as an associated infrastructure, and limited technical aptitude on users of the system led to sabotage and its stalling. The authors suggest that, for a digital system to thrive, proper change management and workforce upskilling and re-skilling must be instituted [21].

Meanwhile, Zeng et al. [24] identified managerial and cultural factors such as information confidentiality and ownership structure of organizations involved in the container booking process, challenging the diffusion of open digital platforms for container bookings.

On the other hand, Nguyen et al. [35] revealed that the TradeLens digital platform, which was developed in 2018 to facilitate seamless documentation formalities in maritime logistics operations, came apart at the seams as it faced the challenge of low critical mass when external stakeholders resisted using the platform. The initiative had been a collaboration between the shipping and tech giants Maersk IBM.

Sakita et al. [23] synthesized 15 challenges that ports face in implementing DT. The authors suggest that one size does not fit all and that research that pays separate attention to challenges specific to individual port organizations is necessary.

Gausdal et al. [4] empirically synthesized the challenges of implementing block chain technology in Norwegian maritime companies; the authors argue that high implementation cost, limited technological diffusion across stakeholders, and risk averseness hinder blockchain diffusion.

Heilig et al. [36] further reveal that DT in ports faces societal challenges in addition to cultural and managerial challenges. The authors argue that port organizations' ability to upskill and reskill their workforce to utilize emerging technologies would facilitate their penetration. Unfortunately, most ports have not yet been able to achieve this as investments in training programs blow up their budgets and increase operating costs [37].

Moreover, Lin [38] argues that for digital innovation like blockchain to thrive, regulatory support is imperative; meanwhile, actors in the maritime industry must deal with the issues of information breaches and exposure to sensitive data. The author suggests that to overcome these insecurities, different layers of information that could be shared across the maritime supply chain can be decided upon upfront, and access can be allowed accordingly, depending on the role each player has in the ecosystem.

Bavassano et al. [31] further revealed that studies on block chain adoption in the maritime industry have largely focused on its technical aspect. The authors argue that the implementation aspects, which largely involve socio-technical considerations as well as legal ramifications, need to be scrutinized. Table 1 below provides a summary of identified DT's challenges in the extant literature.

**Table 1.** Related literature.

| Challenges | Related Literature |
|---|---|
| Digital literacy | Gausdal et al. [4]; Sakita et al. [23], Carlan et al. [15]; Raza et al. [3]; Djoumessi et al. [39]; Gekara and Nguyen [21]; Chowdhury et al. [29]. |
| Cyber security concerns | Nguyen et al. [35]; Lin [38]; Bavassano et al., [31]; Raza et al. [3]. |
| Investment cost | Gausdal et al. [4]; Sakita et al. [23]; Djoumessi et al. [39]; Lambrou et al. [40]; Chowdhury et al. [29]. |
| Digital awareness | Sakita et al. [23]; Gausdal et al. [4]; Philipp [41]. |
| Systems interoperability/incompatibility | Brunila et al. [42]; Nguyen et al. [35]; Carlan et al. [15]; Raza et al. [3]; Inkinen et al. [26]. |

| Challenges | Related Literature |
| --- | --- |
| Culture change | Brunila et al. [42]; Nguyen et al. [35]; Heilig et al. [36]; Zhang and Lam [10]; Gausdal et al. [4]; Raza et al. [3]; Vairetti et al. [28]; Inkinen et al. [26]; Chowdhury et al. [29]. |
| Legal dilemma | Nguyen et al. [35]; Bavassano et al. [31]; Carlan et al. [15]. |
| Lack of policies and regulatory framework | Nguyen et al. [35]; Lambrou et al. [40]; Chowdhury et al. [29]. |
| Environmental uncertainty | Sakita et al. [23]; Nguyen et al. [35]; Carlan et al. [15]. |
| Disinclination to share information | Sakita et al. [23]; Bisogno [43]; Yang [44]; González-Cancelas et al. [45]; Zeng et al. [24]. |
| Executive support | Heilig et al. [36]; Zhang and Lam [10]; Gausdal et al. [4]. |
| Absence of financial support | Carlan et al. [15]; Djoumessi et al. [39]. |

### 2.3. Principal-Agent Theory (PAT) and Ports' DT

In this section, we provide a succinct overview of the PAT theoretical framework and weave through its conceptual understanding of a set of five propositions. Drawing upon this theory, we explore the dynamics between the principle (national government) and the agents (port authority and the LPCO agencies) to uncover the underlying mechanisms that influence the successful implementation of DT in ports in emerging economies. The five propositions, as demonstrated in Figure 1, elucidate the nature of agency relationships and their implications for port governance, stakeholder engagement, and strategic decision-making.

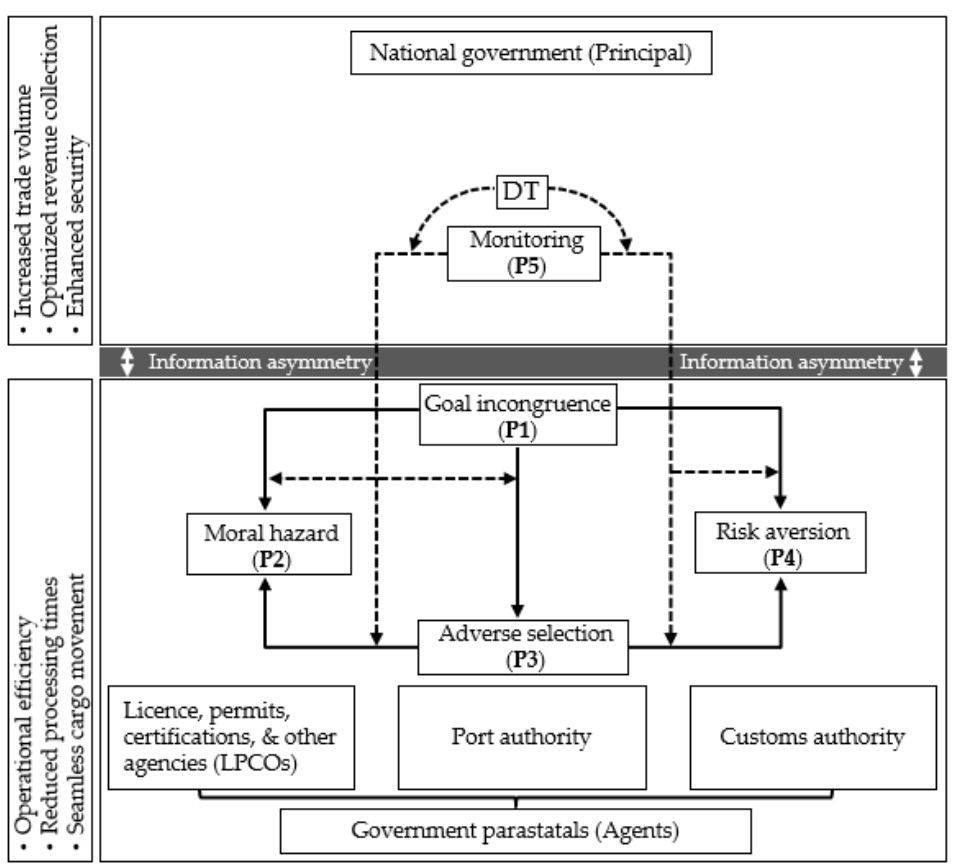

**Figure 1.** PAT framework of DT's challenges in the maritime ports sector.

The PAT, while regarded as one of the most critical theories in finance and economic literature [46], demonstrates limited applicability in the supply chain management discipline [30]. For instance, its applicability in maritime supply chains in explaining contextual

relationships between actors such as national governments and their agencies, such as port authorities, customs authorities, and other public agencies, is lacking. Thus, PAT offers a valuable framework for examining the dynamics within the relationship among national governments as principals and port authorities and other public agencies responsible for trade facilitation at ports as agents. This attempt is one of the early efforts to deploy PAT on a novel phenomenon such as DT in maritime supply chains. Consequently, we derive five propositions using the theory's ramifications in the subsections below.

### 2.3.1. Goal Incongruence and DT

The complex stakeholders' ecosystem in ports reflects varying priorities, which are attenuated by conflicting resource allocation, regulatory disparities, differing performance metrics, and political and bureaucratic complexities [9,47,48]. For instance, national governments typically focus on overarching economic growth and security and cascade these objectives to their agencies, who must interpret them in the confines of their own institutional policies and guidelines. This may create regulatory disparities at institutional levels that may contradict concerted efforts to implement DT initiatives and agencies shirking [47]. Furthermore, as national governments have strict budgetary spend and competing demands, they may incommensurately allocate developmental funds to their agencies, who must then prioritize their spending according to their pressing operational needs. This conflicting resource allocation may lead to insufficient funding for DT initiatives [15]. Worse yet, the different agencies have contrasting performance evaluation metrics that follow the policies and regulations governing their existence. For instance, the port authority may be evaluated based on cargo volume transacted, whereas customs authority is evaluated based on revenue collection efficacies. These diverging performance metrics can lead to situations where either the port authority's or any other agency's success comes at the expense of others' objectives, creating friction and resistance to collaborative efforts [15,49]. Meanwhile, political and bureaucratic complexities may exacerbate power struggles among these agencies and complicate coordination and standardization efforts [47], further aggravating the lack of a unified approach in seaports DT.

**Proposition 1:** *Successful implementation of DT initiatives in seaports is contingent upon the effective alignment of incongruent interests between the national governments and their port-related agencies.*

### 2.3.2. Moral Hazard and DT

A moral hazard situation occurs when there is an information asymmetry between the national government and the port authority or other agencies in the port ecosystem. Typically, the port authorities may possess information about port operations and management outcomes that the government does not have. As the government can only readily observe ports' outcomes but cannot control their actions, the port authority or the other agencies may exploit this fact and behave opportunistically [47,50,51]. This is because it is difficult or expensive for governments to verify what the port authority or the other agencies are doing [32]. The government can institute a penalty or reward (i.e., sacking or promoting agencies' executive directors) after observing some information correlated with or affected by the port's actions [51]. However, such moves lack efficacy as they are performed retrospectively and are not commensurate with the actualization of ill-intended actions. Notwithstanding this, the greater the gap in information advantage between the national governments and the ports, the higher the likelihood of moral hazard prevailing and the lower the chance of successful implementation of DT. While DT initiatives can arguably dampen information asymmetry and the associated moral hazard of the agencies, the prevalent inclination to extract private rents subsumes such initiatives.

**Proposition 2:** *Inclination to misappropriate illicit rents by port authorities constrains ports' DT endeavors due to perceived transparency and accountability digital systems enforce.*

### 2.3.3. Adverse Selection and DT

Adverse selection emerges when the principal lacks comprehensive information about the actions and intentions of the agent, leading to suboptimal outcomes [50]. Typically, national governments seek to improve trade facilitation, security, and revenue collection and appoint agency leaders who will help achieve these objectives. However, the port-related agencies possess a nuanced understanding of the operational practices and challenges of the port environment, which may not reflect the national government's objectives. This privy information would be pertinent to the governments' decisions in favor of their own interests [51]. The national government may not be fully informed of the extent of manual interventions and their implications on port operations and stakeholder interactions. This information asymmetry stifles visibility [3] and can result in the port authorities undertaking actions that do not serve the best interests of the government (i.e., agency loss) [51,52]. Furthermore, it could result in inefficient resource allocation if the government allocates funding based on a lopsided understanding of the actual needs and conditions of the port authority and the agencies. We argue that adverse selection concerns emerge when the port authority does not take steps to address these inefficiencies and modernize its operations.

**Proposition 3:** *Port authorities whose top leaders are not keen to streamline their processes demonstrate limited DT footprint.*

### 2.3.4. Risk Aversion

Risk aversion arises when the principal and the agent display disparate levels of risk tolerance [48], which determine the course of action each takes in a given situation. For instance, when employed by the national government, the agents are unable to diversify their employment, while principals can diversify their investments [32]. This imbalance in risk distribution makes agents assumedly risk averse and likely to shirk [47]. Hence, we argue that while the governments may readily advance investment funds for digital solutions, the port authorities and related agencies may demonstrate risk averseness. This may stem from perceived potential disruption of the status quo and inherent operational and technical risks associated with such investments. For example, the port's workforce may be highly old and inept, and this may present inertia towards the implementation of digital solutions [4,21]. Meanwhile, technical expertise may be lacking, and thus, agents may shun implementing digital initiatives. As Wang et al. [53] posit, risk-related behaviors significantly impact port authorities' decisions relating to long-term investments. As digital technologies evolve rapidly, they present uncertainty [32,54], which ports may shun by expressing inertia to invest in them. The fear of failure and possible repercussions from a wasteful commitment to dire governments' budgets complicates risk averseness.

**Proposition 4:** *Different risk perceptions between national governments and port-related agencies attenuate disinclination to the risk appetite it takes to promote DT footprint in seaports.*

### 2.3.5. Monitoring and DT

The proponents of principal-agent theory posit that agents are inherently self-interested and inclined to pursue their own objectives at the expense of their principal [47,50]. Thus, for the principals to ensure that agents live up to their expectations and avoid agency shirking, they institute monitoring and controlling mechanisms [32,47,48]. From the principal's perspective, ports' and agencies' DT will overcome the problem of information asymmetry, which aggravates moral hazard and adverse selection [32]. It will thus attenuate agencies' costly monitoring [47,52]. For instance, the extant literature argues that DT facilitates process transparency and visibility, leaving behind an audit trail that can be used in some form of agency monitoring tool [3]. Consequently, the principal can garner the true picture of agents' activities and processes and further develop incentives and sanctions that entice the agents to align their goals with those of the principal.

**Proposition 5:** *Successful implementation of DT in ports will enhance principals' monitoring efficacy in reducing agency costs; however, this relationship is contingent upon the nature of principals' monitoring practices.*

Thus, Figure 1. illustrates the proposed maritime port's DT framework through a P-A theoretical lens.

**3. Research Methods**

Because DT is quite nascent in seaports, we used a case study strategy and abductive approach to develop an in-depth understanding of its tenets and analyze contextual events and their relationships in accordance with Yin [55]. The abductive approach allowed us to apply propositions to the empirical data Kovács and Spens [56]. While qualitative methods are generally suited to theory development, they can also be applied in deductive reasoning in what Kovács and Spens [56] term theory matching. Consequently, we examined Port Omega's DT efforts by interviewing senior officials regarding their lived experiences and interactions with the port. Based on the understanding that ports are slow in adopting DT [57], our objective has been to empirically explore the challenges of undertaking the successful implementation of DT in maritime ports by drawing evidence from a developing country. Following an extensive literature review, we developed and furnished the interview protocol (Appendix A) with open-ended questions that were meant to capture the DT state of affairs at Port Omega. The sampling procedure included other Port Omega's value chain actors such as customs authority; clearing and forwarding agents; shipping agents; agencies dealing with licensing, permits, certificates, and others (LPCOs); traders; and shippers council representatives. The inclusion of these actors enabled us to collect additional triangulation data by cross-referencing stakeholders' perspectives, building extra heterogeneity into the sample, as suggested by Robinson [58]. This engendered a thorough understanding of DT trajectories and nuances from multiple perspectives and reinforced the robustness, validity, and transferability of the underlying results.

With respect to the transparency of the research process, we resorted to using pseudonyms to disguise the real names of the port organization and the stakeholders from which we elicited information. In this regard, we use "Port Omega" to protect the privacy and integrity of the stakeholders involved. This has been particularly integral to upholding our ethical obligations as both the real word case port and its value chain stakeholders did not offer us prior permission to reveal their real identities. In the process of negotiating access to these respondents, we promised them that any identifiers would subsequently be removed from the data when they explicitly expressed their concerns about the revelation of their true identities. Moreover, stark verbatim quotes offered by interviewees, particularly those who are internal to the case port, may jeopardize these individuals and lead to unintended consequences such as reprimand and social prejudice [59,60]. The authors have been alerted that revealing the true identity of Port Omega could potentially give away the identities of respondents whose personal information, such as positions and roles, have been explicitly presented in Table 2. Therefore, it has been our duty to ensure ethical consideration in light of all stakeholders we recruited in our sample. Nevertheless, studies that deploy pseudonyms instead of real names of people or places are not uncommon in the extant literature, as data protection rights and requirements for ethical considerations in conducting research are on the rise. See, for instance, Heaton [60]. Notwithstanding this limitation, the researchers have otherwise adhered to high research standards, as evidenced in the systematic identification of the problem, case protocol development, data analytical procedures, and synthesis of results. The entire research process is thus traceable and replicable and unravels DT in a port context both constructively and objectively [61].

**Table 2.** Information of interviews.

| Organization ID. | No. of Interviews | Interviewee's Role | Interview Duration (Mins) | Role in Port Value Chain | Work Experience (Years) |
|---|---|---|---|---|---|
| SM1–PA | 1 | Senior Manager IT1 | 40 | Port authority | 7 |
| OM–PA | 1 | Operation Manager | 53 | Port authority | 11 |
| SM2–PA | 1 | Senior Manager IT2 | 85 | Port authority | 9 |
| OM–STV | 1 | Operation Manager | 45 | Stevedore | 14 |
| CM–CA | 1 | Customs Manager | 68 | Customs authority | 7 |
| BO–CAG | 1 | Business Owner | 70 | Customs agent | 7 |
| SO–FFA | 1 | Senior Officer | 74 | Clearing and forwarding Agents' Association | 17 |
| SO–SAA | 1 | Senior Officer | 66 | Shipping Agents' Association | 13 |
| OM–PAREG | 1 | Operation Manager | 119 | Regulator of the port authority | 10 |
| CM–QREG | 1 | Quality Assurance and Compliance Manager | 126 | Regulator | 8 |
| OM–TRD1 | 1 | Operation Manager | 62 | Trader | 6 |
| LM–TRD2 | 1 | Logistics Manager | 50 | Trader | 12 |
| SO–SHC | 1 | Senior Officer | 58 | Shippers' Council | 18 |
| Total | 13 | | 916 | | |

### 3.1. Case Description

Port Omega is a state-owned parastatal that is strategically located along the stretch of the Indian Ocean's coastline that borders an emerging economy country on the east side. This position poises the port as a pivotal trade-facilitating hub serving several landlocked neighboring countries. The port handles about 90 percent of international freight and depicts an average growth of 6.4 percent annually. In 2021, Port Omega registered about 16 million tons of throughput cargo it serviced. Port Omega has increasingly become a center stage for economic development in East and Central African regions. Such centrality has attracted immense attention and pressure from both the government and stakeholders on the port to fortify its operational and economic efficiencies. This has seen numerous developmental projects underway, such as the port's berth expansion, dredging, and investments in new digital systems such as cargo x-ray scanners, integrated electronic payment systems, and eSWS. These technological advancements do not sufficiently parallel the ever-growing cargo volumes transiting through Port Omega's frontiers as investments in digitalization projects have almost consistently faltered. Port Omega operates both as a landlord and service port. It has leased part of its berths to a private terminal operating company that solely handles containerized cargo. Likewise, Port Omega handles all sorts of cargo (i.e., general, containerized, etc.), which complicates its operations given the numerous interactions it has with other stakeholders. Thus, Port Omega becomes a suitable case for investigating DT in the emerging economy context.

### 3.2. Data Collection

Our data collection process unfolded through three specific interrelated facets: (1) determining the cases to be included in the sampling frame, (2) determining the sampling strategy, and (3) performing the actual data collection. Bearing in mind the quest to explore the challenges facing ports in light of undertaking DT initiatives, the port authority (Port Omega) had been our initial prime target. This was because port authorities, in many instances, are considered pioneers who orchestrate DT initiatives in their ecosystems [23]. Yet, in recognizing the fact that port authorities do not exist in isolation and that their actions and efficacies rely on other pertinent stakeholders [25], we deemed it imperative to also recruit the latter. The logic was such that other stakeholders external to the port authority would provide many objective perspectives about the port authority's DT endeavors, which might not have been provided by the latter's internal stakeholders. Consequently,

the perspectives from multiple stakeholder groups, as detailed in Table 2, not only afforded us a much more comprehensive understanding of the DT phenomenon in Port Omega but also added a layer of heterogeneity into our sample and fostered the validity of our results [55].

Having decided on the cases, we went on to determine the sampling strategy. The researchers favored two non-probabilistic strategies: purposive and snowballing [3,55]. The purposive strategy has been appropriate as the respondents whom we sought to recruit in our sample had to meet a minimum set of requirements such as seniority, information richness [62], and the role played in the port ecosystem's value creation. Moreover, following our choice to include other pertinent stakeholders in the sample, we used the snowball strategy to expand the sample. Here, the researchers asked the respondents at the end of each interview to identify actors whose actions and interactions had a significant bearing on Port Omega's operations. We used this as a proxy to assess the relevance of additional samples we recruited all along.

The actual data collection occurred between May and July 2022. Using the sampling strategies highlighted in the preceding paragraph, we conducted a total of 13 semi-structured interviews with Port Omega' and stakeholders' senior managers. The use of semi-structured interviews allowed us to wage a guided interview process while incorporating emerging views as port stakeholders opened out about their lived experiences with regard to Port Omega's DT endeavors. Categorically, the interview process included 10 different stakeholder groups, as highlighted in Table 2. For instance, Port Omega, abbreviated as (PA) had three respondents, while other stakeholders had one representative each. The interviews were either in-person, through telephone, or on Microsoft teams. All interviews were audio-recorded to abate errors in recounting respondents' narratives, minimize interpretive bias, and safeguard reliability [55,63]. On average, the interviews lasted a little over one hour. Moreover, the field notes and Port Omega's walkthrough provided us with valuable information that shaped our narratives. Similarly, immersion into the secondary sources of data, such as reviewing Port Omega's handbook and the chief auditor general's reports, among others, engendered the researcher's further emic understanding of the research context.

*3.3. Data Analysis*

In performing data analysis, the researchers adopted the following methodological approach: (1) transcription of audio-recorded interviews, (2) immersion of the transcribed interviews, and (3) the coding process. This structured approach demonstrates analytical rigor aimed at ensuring the integrity and reliability of the findings.

Regarding the transcription process, the interviews conducted both in the local language and English were carefully transcribed accordingly to preserve the pristine nature of the data. This practice not only upheld the validity and accuracy of the information gathered but also mitigated the potential interpretative biases that might have slipped through due to researchers' subjective interpretations. Nevertheless, the latter is ineluctable [64] in qualitative research, where researchers are instruments for data collection [65]. The decision to translate quotes from the local language to English further enhanced coherence and facilitated the seamless integration of the data into subsequent analysis.

Prior to initiating the coding process, we immersed ourselves in the transcripts by intensively reading through them in accordance with Braun and Clarke [66]. This immersion provided the researchers with threefold opportunities. Firstly, we were able to develop a comprehensive understanding of stakeholders' perspectives, insights, and viewpoints regarding DT in Port Omega. Secondly, it allowed us to familiarize ourselves with subtle nuances and intricacies embedded in the data. In fact, the immersion helped us discern recurring themes and gain insights into Port Omega's context. Thirdly, it facilitated the identification of relevant segments of key quotes that encapsulated vital ideas shared by the respondents, thereby enabling us to streamline the data and highlight sections that warranted closer scrutiny during the coding process. Thus,

this process of familiarization with the transcripts was pivotal in establishing a solid foundation for the subsequent analytical stages.

Then, we augmented the transcripts with corresponding field notes we took during the interviews. Better yet, the secondary sources facilitated reflexive narratives as we charted through results and discussions. Moreover, the walkthrough conducted by one of the researchers enhanced our insights into the port's operational environment. The insights further helped the researchers enrich the interview process as more respondents were recruited into our sample. Both the walkthrough and additional secondary sources enabled us to corroborate the evidence that emerged from the interviews, thus offering a more balanced narrative of the state of affairs of the DT phenomenon in Port Omega.

Next was the coding process that ensued in NVivo Software V20.1. The use of NVivo analytical tool allowed us to structure, organize, and analyze about 156 pages of transcribed qualitative data. Following the prescribed methodology outlined by Gioia et al. [67], we embarked on a three-stage coding process: (1) first-order inductive coding in stage 1; (2) second-order code categories in stage 2; and (3) themes generation in stage three. These stages and the synthesis of data are encapsulated in Figure 2.

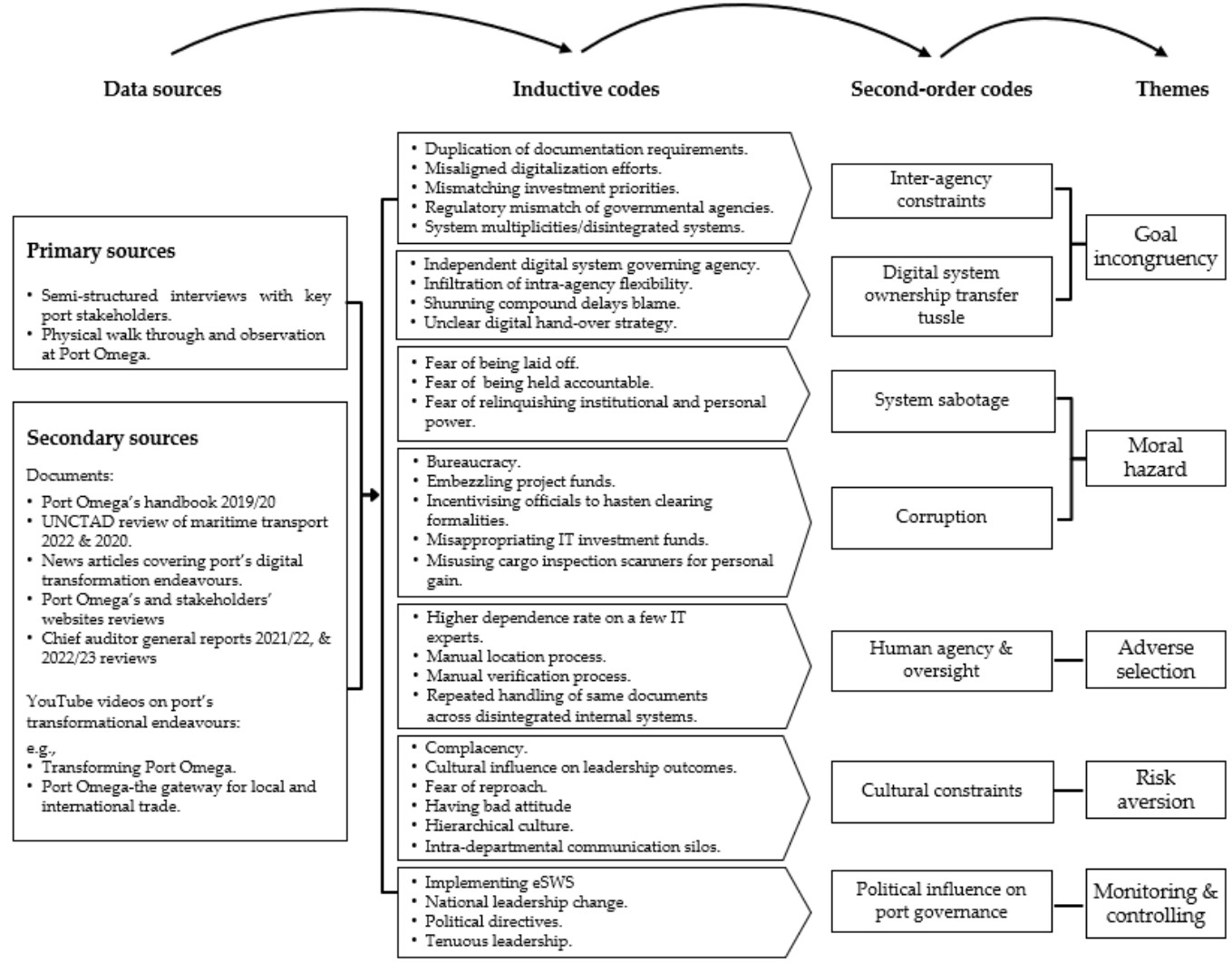

**Figure 2.** Analytical procedure.

The first-order coding process enabled us to develop inductive-verbatim codes, meaning that the researchers recorded important concepts that emerged from the dataset as data analysis unfolded. This has been particularly important to ensure the pristine nature of the

respondents' viewpoints without necessarily infiltrating our own preconceived notions. As pictured in Figure 2, the inductive codes represent aspects that are reflective of the underlying meanings, concepts, or themes in empirical data and particularly preserve the richness and depth of the participants' experiences, perspectives, and insights. Eventually, we developed 29 inductive codes in this stage.

Further down the analytic process, we synthesized and construed meanings of the initial codes into stage 2's code categories or second-order themes as referred to by Gioia et al. [67]. Here, the level of granularity of empirical data waned as the researchers introduced concepts from the extant literature in an attempt to encapsulate the initial codes. This process of developing code categories involved the researchers in iterative processes of juggling between the dataset and already existing empirical concepts in what Saldaña [63] terms abductive coding, meaning that which combines inductive and deductive logic [67,68]. Consequently, the deployment of the code-to-theory approach enabled us to further shrink the 29 codes into 7 s order codes in the third column in Figure 2.

In stage three, we conceived the code categories into five PAT's aggregate themes in column 4 in Figure 2. We then reached a consensus on aggregate themes and reconciled any disagreements that arose during the coding process. This alignment underscores the collaborative nature of the analysis and reflects the researcher's commitment to rigor and transparency. Of note, the level of abstraction increases as data assimilation tapers down. This is an important element of theory building [43], even though this was not the aim of this study regarding theory elaboration. Notwithstanding this, the iterative nature of the coding process has allowed us to refine and validate the codes through ongoing engagement with data. As we progressed from first-order inductive codes to aggregate themes, we continuously compared, contrasted, and revised the codes to ensure their accuracy and relevancy in capturing the essence of the stakeholders' contributions.

## 4. Results

In this section, this study presents qualitative findings derived from the investigation into the dynamics of implementing DT in maritime ports through the lens of PAT. This study sheds light on the intricate relationships and power dynamics that characterize interactions between the national government as the "principal" and port authorities and other agencies as "agents".

### 4.1. Descriptive Statistics of Respondents' Profiles

The 13 interviews involved three of Port Omega's internal stakeholders, who formed 23.1% of all respondents. This had been logical as the port had been the unit of analysis. The remainder, 76.9%, constituted nine categories of respondents external to Port Omega. These external stakeholders had been pertinent value chain actors who provided critical triangulation perspectives. Of all respondents, 61.5% constituted males, while 38.5% constituted females. Meanwhile, the average work experience of the respondents had been 10.7 years. This meant that we garnered rich verbal information from experienced port ecosystem stakeholders. Similarly, 81.5% of all respondents contributed, on average, to the formation of the five themes. Table 3 illustrates the frequency distribution of themes and supporting evidence. Extended details can be found in Appendix B.

In the next five subsections, we thematize the challenges, provide adequate direct evidence of our empirical observations, and artfully integrate this evidence into narratives with the aim of honing the clarity and impact of our findings.

**Table 3.** Frequency distribution of themes and supporting evidence.

| No. of Inductive Codes | Theme | SM1–PA | OM–PA | SM2–PA | OM–STV | CM–CA | BO–CAG | SO–FFA | SO–SAA | OM–PAREG | CM–QREG | OM–TRD1 | LM–TRD2 | SO–SHC | (f) | % |
|---|---|---|---|---|---|---|---|---|---|---|---|---|---|---|---|---|
| | | | Internal | | | | | | | External | | | | | | |
| 9 (Ref. Figure 2.) | Goal incongruence | | (2) 17% | | | | | | | (10) 83% | | | | | 12 | 92.3 |
| 8 (Ref. Figure 2.) | Moral hazard | | (2) 20% | | | | | | | (8) 80% | | | | | 10 | 76.9 |
| 4 (Ref. Figure 2.) | Adverse selection | | (2) 18% | | | | | | | (9) 82% | | | | | 11 | 84.6 |
| 6 (Ref. Figure 2.) | Risk aversion | | (3) 30% | | | | | | | (7) 70% | | | | | 10 | 76.9 |
| 4 (Ref. Figure 2.) | Monitoring and controlling | | (1) 10% | | | | | | | (9) 90% | | | | | 10 | 76.9 |

*4.2. Goal Incongruence*

The issue of goal incongruence emanated from digital orchestration challenges between the two powerful stakeholders in Port Omega's ecosystem: Port Omega itself and the customs authority. The generation of this theme had been supported by 92.3% of all respondents, of which 17% were Port Omega's internal staff and 83% Port Omega's external value chain actors like the regulator, customs authority, and customs agent who have closely linked government agencies with a mandate over imports and exports. Thus, our results revealed that inter-agency constraints emanated from agencies' actions and interactions, which conflict with concerted efforts to undertake DT endeavors. For instance, 23.1% (See Appendix B) of the respondents claimed that the LPCOs' disparate performance appraisal indicators compelled them to either aggressively engage in investments in DT initiatives or divert investment resources towards other internal developmental projects like upgrading physical infrastructure. When asked what they thought about the slow DT processes in Port Omega's ecosystem, the respondents highlighted that the port authority (Port Omega), for instance, had been concerned with attracting more cargo volume and improving physical infrastructure, whereas the customs authority had been concerned with revenue collection efficiencies. Consequently, the latter had been involved in investing in internal digital systems to that end, creating disparities that defeat unified DT orchestrations. A customs manager affirmed that:

> [...] As for customs authority, we are required to generate revenue that supports the running of the country, we are the eye of the government, most often facing the direct pressure to ensure revenue collection sustainability, as such we are constantly engaged in digitalizing our systems. Meanwhile, the port authority has a different performance appraisal criterion, it is only required to submit dividends which depend on the profit it has earned during an accounting period, this may discourage port's digitalization efforts as an avenue to streamline its operations, the port authority does not have an incentive to budge.
>
> (CM–CA)

Furthermore, our results revealed the existence of a regulatory mismatch among governmental agencies, which the respondents claimed had a bearing on digital innovation initiatives as these stakeholders displayed inconsistent implementation of DT initiatives. The respondents consented that Port Omega's regulator lacked efficacy and relevance in the ecosystem as it cannot exert the same coercive pressure on all external stakeholders. This situation heightens disparities in transformation trajectories across actors and the overall uncompetitiveness of the port, as a manager asserted:

> [...] We closely monitor and regulate Port Omega's activities, furnish it with improvement suggestions and ensure their implementations ... nevertheless, it is not the port authority's sole responsibility to facilitate efficient digital platforms. Therefore we [the regulator] will certainly compel the port authority to implement and use a digital platform whether they like it or not but, we cannot make other LPCOs do the same because we are not mandated to do so.
>
> (OM–PAREG)

About 53.8% of all respondents also indicated that Port Omega's ecosystem can hardly ever have a common system that adequately fulfills the requirements of each stakeholder pertinent to the value creation equation. For instance, by design, Port Omega and other agencies (LPCOs) have their own disparate regulations and policies that govern the way they manage and conduct their business. As such, the respondents argued that having common ground on digitalization endeavors becomes a futile affair. A senior officer in the shipping agent association added the following:

> [...] Port Omega's stakeholders do not and cannot have a single common system because of the regulations and laws that govern their existence. Disparate sys-

tems are imperative for facilitating operations, formalities, and procedures that distinguish each stakeholder.

(SO–SAA)

An important finding that characterized the scrambling of stakeholders when their interests are at stake involved the tussle between the port authority and customs authority for the ownership and control of an eSWS. About 30.8% of respondents concurred that the tussle emanated from the belief that both Port Omega and the customs authority play a major role in the ecosystem and, therefore, should have full control over the eSWS. Meanwhile, the port regulator contended that should the eSWS be owned by either Port Omega or the customs authority, one of them will lose the flexibility it takes to provide port-related services and thus unnecessarily assume blame that could have been avoided had the system been hosted in-house. A shipping agent manager further added that, as the port authority and customs authority vie for eSWS ownership, each of them refrains from wholly furnishing the system's developers with all requisite user requirements that could help optimize it. Consequently, actors have resorted to using their legacy systems alongside an eSWS, defeating its efficacy in fostering transparency and accountability. Port Omega's IT manager affirmed that:

> [...] We have been having a difficult time agreeing on who should take ownership of the eSWS, phase one which was meant to integrate all the LPCOs has been completed however, we want to host the system as port authority, and so does the customs authority. As we speak the eSWS has not been handed over to any of us because we think we will be rendered ineffective in our handling of port services as much as the customs authority does ... I think we fear relinquishing the power that comes with hosting the system.
>
> (SM2–PA)

### 4.3. Moral Hazard

This theme related to the challenge of undertaking DT that emanated from the prevalent opportunistic expropriation tendencies in Port Omega's ecosystem. As indicated in Table 3 and Figure 3, 76.9% of all respondents, 20% of whom were internal stakeholders, while 80% were external stakeholders, provided perspectives that indicated how corruption and intentional system sabotage had stifled DT initiatives from taking root over the years. This sentiment is largely shared by external stakeholders who are the prime users of Port Omega's services. For instance, a senior officer in shipping services revealed that investments in digital solutions would likely eliminate the elusive activities by port officials who often connive with external stakeholders in such acts as a spurious declaration of cargo details for their own personal gain. Our findings revealed that as digital technologies are hailed for enhancing transparency and visibility of port processes, DT thus perpetrates perceived fear on the part of some officials who, in turn, sabotage the implementation of DT initiatives in order to continue illicit expropriation of rents only possible through manual interventions.

> [...] You know, systems eliminate bureaucracy ... and through bureaucracy, people earn their extra [illicit] income. Now, if systems operate efficiently, not every employee likes it in their workplace. This is because the systems will deprive them of something, prevent them from meeting certain people [when they can expropriate private rents], or even make them anonymous in certain places.
>
> (SO–SAA)

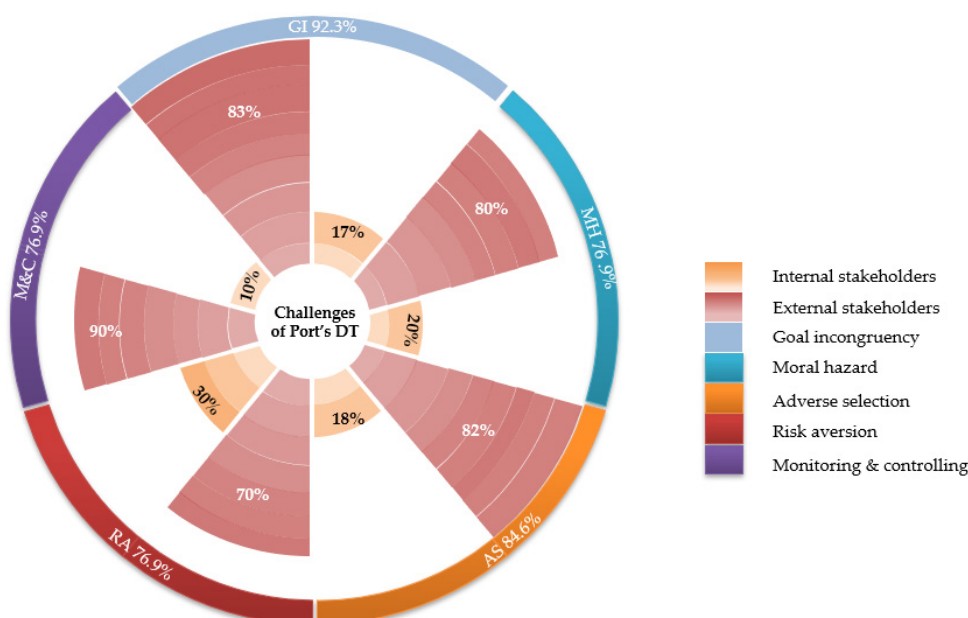

**Figure 3.** Percentage distribution of empirical evidence of challenges of undertaking DT in maritime ports in emerging economy context.

From an internal perspective, Port Omega's workforce also affirmed that they had experienced considerable inertia to obtain digitalization initiatives to materialize; they argued that if it had not been embezzlement of digitalization funds, then it was a group of users boycotting implemented systems as they were excluded from onboarding training abroad which had come with monetary incentives. Port Omega's IT manager further affirmed these concerns:

> [...] We have encountered a number of challenges getting digitalization initiatives to materialize at the port, ...over the past decade we aimed to undertake several ICT-related investments but fell short and only a few fully materialized. For instance, we wanted to implement an ERP [enterprise resource planning] system to streamline port operations and departmental coordination, it was a very big project, but there were a lot of problems as it was not going live since 2017, and still has not gone live as we speak ...I can't speak of everything but, ...there is sabotage going on here. The same happened to eSWS that was meant to go live since 2015, it never did until recently when the government pulled the project and assigned a local system developer who also encountered a lot of resistance and dissent. ...I think it has been due to the fear that the systems would remove the loopholes of fraudulent rents exploited by officials.
>
> (SM1–PA)

Furthermore, our findings revealed that while the government insisted on the acquisition and implementation of digital solutions in Port Omega's ecosystem, the exchange of digital documents had been perceived to eliminate the necessity for in-person visits to Port Omega and the LPCOs and consequent opportunities for collusion. Moreover, according to respondents external to the port authority, the manual handling of physical documents involved face-to-face interactions, which had been exploited for corrupt purposes. Both petty corruption and kickbacks have been reported to be Port Omega's business-as-usual culture, and port officials deliberately attempt to stall DT initiatives as they are aware that their actions would otherwise be traceable. In this vein, a trading manager among 23.1% of others asserted that:

> [...] The online transmission of documents has got transparency ...if you visit the port, face-to-face discussions generate other unnecessary discussions and the

room for being asked for a small favor to incentivize someone to work, …but even with the digital system in place, the port's officials sometimes deliberately sit on the documents by claiming the systems to be out to entice port users to visit in-person when they get asked for corruption.

(OM–TRD1)

The respondents further affirmed that the lack of transparent and integrated systems at Port Omega had exacerbated illicit acts of misappropriation that had been the motivation for some officials to sabotage investments in DT. For instance, the prevalent manual exchange of documents in the ecosystem had compelled the port users to offer kickbacks to get things done, whether clearing cargo or misrepresenting consignment details to avoid paying commensurate tax, as a respondent posited:

[…] You lodge clearing documents online however, you still have to go physically to push them, and this is where bribery arises because such physical contacts entice you to offer a small token …this has become a culture and I think only transparent and streamlined systems can overcome it.

(BO–CAG)

Our findings also revealed that, besides the petty corruption and shady deals, there had also been a bigger problem in Port Omega. For instance, the respondents revealed an insidious embezzlement of digitalization-related investment funds and associated it with the port's large-scale corruption. They argued that the deliberate misappropriation and diversion of funds allocated for digital infrastructure development had undermined the integrity of financial systems and struck at the heart of efforts to modernize Port Omega's operations through DT. This sentiment had been well echoed by a senior official who affirmed that:

[…] Port Omega embarked on almost a \$6.7 million project that was meant to bring about an eSWS more than half a decade ago, the project did not actualize because the funds had been misappropriated and embezzled at institutional level, …later on the project was re-initiated, somewhere down the line, it again became tainted with a corruption scandal, so you can get a picture of the environment DT must thrive in.

(SO–FFA)

Worse yet, our findings revealed that despite the potential benefits of integration, the prevailing trend in Port Omega's ecosystem had been that of maintaining disintegrated systems. The respondents cited concerns over corruption as the primary reason for this fragmentation. They expressed apprehension that a centralized system would limit their discretion and potentially expose corrupt practices. Similarly, the respondents perceived the siloed institutional systems to be a means of retaining control over port users. For instance, they argued that by maintaining disparate systems, stakeholders can exert influence and manipulate processes to their advantage. Accordingly, the decentralization of systems has not only been believed to perpetuate a culture of corruption but also hamper DT efforts, which target the streamlining of port ecosystem operations and enhancing greater transparency. This narrative is supported by a quote from a seasoned shippers' council representative (SO–SHC) who commented that:

[…] Each of the major stakeholders in the port has been developing their own institutional systems, this one is developing its own system for tax collections, that one is developing theirs for cargo clearing purposes, another is developing their own system for quality control, but there is not a central software that integrates these stakeholders under a common system. Now, each of these stresses on maintaining disintegrated systems due to corruption, they know once they have a centralized system, they will lose the discretion to manipulate port users.

(SO–SHC)

*4.4. Adverse Selection*

This theme concerned agencies' actions that contradict the principal's intention of digitalizing the port ecosystem. The key attributes uncovered by our study had been a higher dependence rate on a few IT experts, manual location and verification processes, and duplicate processes among agencies. About 84.6% of all respondents, 18% of whom were Port Omega's internal staff and 82% external stakeholders, contributed to the formation of this theme.

Our findings revealed that Port Omega relied heavily on a limited number of IT experts for DT initiatives. This dependence has raised issues regarding the ability of the port's workforce to provide the expertise it takes to effectuate DT. The findings further revealed that when a few IT experts hold significant influence over digital projects, they risk impeding the effectiveness and inclusiveness of digital solutions on the strength of their biases or knowledge limitations. A total of 15.4% of respondents further added that the concentration of expertise among a select few resulted in bottlenecks, delays, and vulnerabilities in the implementation and maintenance of digital systems, thereby exacerbating the challenges faced during the transformation process. A customs authority manager provided their views on this matter, as indicated in the verbatim quote below:

> […] You know we [the port ecosystem] have a few IT experts who can develop and manage those digital systems, …so if a system-related problem arises it disrupts a host of operations until a specific expert personnel comes to rescue. The question is why should we have such a high dependence rate on one person? In my opinion the dependency rate per IT expert should be commensurate with IT usage requirements and not wholly rest with a few experts who if are not there you can clearly see that things are not moving, …now, that dependence rate on a few IT staff in government parastatals is very high.
>
> (CM–CA)

Similarly, our findings uncovered the vulnerability of digital systems to manipulation and exploitation by a few individuals with expert knowledge. By exploiting their position and understanding of the system's intricacies, such individuals have been able to undermine the port's objectives for their own personal gain. This revelation is well narrated by a trading manager who affirmed that:

> […] There was an incident where we were tracking some information and we were told that we could not do that in the [Port Omega's] system, we needed to do it manually, …we came to learn that the person who was handling the system had stepped out, but before doing that they corrupted the system which could not be restored immediately, …so, maybe that is why anyone who is coming up with new systems should look into those loopholes.
>
> (LM–TRD2)

A total of 53.8% of respondents indicated that Port Omega significantly embraces manual interventions. This reliance on manual methods reflects the persistent embracement of outdated practices despite the availability of digital alternatives. For instance, the respondents argued that manual processes not only impeded efficiency but also increased the likelihood of errors, delays, and inconsistencies in the port's value-capturing and transferring processes. They argued that these practices precluded the visibility of processes and necessitated unnecessary and costly duplication of efforts. As in a vicious cycle, the problems of adverse selection and moral hazard reinforce each other to weaken DT diffusion in the port ecosystem.

Additionally, our findings revealed that certain practices were deeply entrenched in the daily routines of key personnel in Port Omega. For instance, one of the port operations managers revealed a longstanding tradition of manual verification of tally sheets during cargo offloading within the container department. This method, though seemingly outdated in the age of digital automation, has remained a persistent procedure within

the department's workflow. Similarly, the stevedore manager corroborated this practice, emphasizing its prevalence in the port ecosystem. Thus, 30.8% of respondents argued that the reliance on manual verification of tally sheets implied a certain level of trust placed in this traditional method despite the potential for errors and inefficiencies. The following quotes provide corroborative evidence:

> [...] The intention was that when unloading cargo, it ought to automatically be tallied in the system. But, because the system has not been functioning properly, we manually record using tally sheets. This is done for all incoming cargo. So, you may find a clerk working with a bunch of paper tally sheets of up to forty pages, ...and the same work is repeated more than once, in our cargo system [Port Omega's], and in the customs integrated system [customs authority].
>
> (OM–PA)

> [...] And it is not that the system cannot be fixed, but it is not working because it is for the benefit of a few people.
>
> (LM–TRD2)

> [...] Yard positioning is overall performed manually. What happens is, a clerk goes to the yard and records information on the location of the container or car on paper called movement sheet, and later on feeds it into cargo system, garbage-in-garbage-out,...a clerk can say they positioned cargo on block B, only to find out after a lot of bother, they are on block D. Unlike the port, the terminal operator [private] has digitalized its cargo system and yard positioning is integrated with scanning devices that eliminate manual positioning and location of cargo. Their system also eliminates manualness when offloaded cargo can be tallied simultaneously with cargo system and updated on customs authority's system ...we have been hoping for a system like that, but that day has not yet come.
>
> (OM–STV)

Moreover, our findings showed that manual transferring, rectifying, and editing of data entries, often conducted on Excel sheets, presented significant vulnerabilities to the port's efficiency and accuracy. These manual processes have been prone to errors, inconsistencies, and redundancies, which have been reported to have compromised the integrity of data and hindered decision-making processes. An operation manager added that:

> [...] We have a problem with the current system ...it can receive manually inputted duplicate data without notifying the user that they have already performed a certain action. So, when producing weekly or monthly management reports, you may find that the figures are spurious and misleading, the system could not detect that, it is a garbage-in, garbage-out kind of system.
>
> (OM–PA)

### 4.5. Risk Aversion

The risk aversion theme encompassed cultural aspects with a bearing on the successful implementation of DT initiatives. For instance, our analysis uncovered complacency, cultural influence on port's leadership outcomes, fear of reproach, hierarchical culture, and intra-departmental communication silos as critical impediments to the diffusion of DT in Port Omega's ecosystem. As depicted in Table 3 and Figure 3, 76.9% of respondents, of whom 30% were internal stakeholders and 70% were external stakeholders, contributed to the formation of this theme.

Our findings unveiled the prevailing negative perception of government parastatals like Port Omega, which have been associated with disinclination to risk-taking. A total of 53.8% of the respondents asserted that such entities have often been characterized by perceptions of sluggishness, an aging workforce, limited transparency, and susceptibility to fraudulent practices. These aspects were argued to contribute to a culture of resistance

to change, which has posed significant challenges to efforts aimed at fostering innovation in such parastatals. For instance, the susceptibility to fraudulent practices determines investment priorities, where those investments that are more prone to embezzlement receive top priorities as affirmed by a senior manager in freight forwarding services and customs authority manager:

> [...] A lot of investment decision in governmental parastatals are made on the basis of the possibility of retaining a clearly identifiable 10 per cent of the procurement cost ... now, investment in digital systems is not something that a lot of people know how they can rip off the 10 per cent. But in procurement of reach stackers, one may claim they also procured 42 spare tires meanwhile they only did 2.
>
> (SO–FFA)

> [...] I think Port Omega's leadership does not prioritize DT in the meantime, ... thus, there is lack of digital technologies and appropriate staff who can push digitalization agenda. Existing staff stay sedentary in one duty station until they grow old, become accustomed to their jobs, and so inert to digital innovations. For instance, the newly appointed port's director general seems to have a vision that could change the port, however the problem is his subordinates are the same old folks, so, what can he change then?
>
> (CM–CA)

Yet, the culture of complacency perpetrates a dichotomy between the familiar and the unknown, hindering the port's ability to fully integrate digital technologies into its operations. Furthermore, the emphasis on individual gains over port-wide collective benefits underscored a myopic view of the value of DT in the port. The fixation on personal benefits engenders resistance to change as the workforce perceives DT as a threat to their established status quo. Similarly, the skepticism towards the government's digital initiatives underscored broader issues of institutional trust and perception, which further complicate efforts to drive DT, as echoed by a trading manager:

> [...] If you go to the port most people are senior, they have been there for ages to the point that some of them even think that they own it, ... besides, there is this notion that if it is [an initiative] government's, then we are still in transition; we are 50 per cent in the system, and 50 per cent out. I think they should understand the value of implementing this DT in the country. And value should not be as per the individual, because you will see people will be asking, as a person what do I get?
>
> (LM–TRD2)

Our findings also revealed what we term a "suck up effect", where the influence or effectiveness of one competent individual within the port ecosystem is limited in effecting positive change or improvement among a larger group of less competent or underperforming personnel. This aspect underscored the challenge of a single individual, no matter how skilled or competent, being unable to significantly alter the overall culture of the workforce characterized by negative traits such as inertia and reluctance to change. The many individuals eventually exert a negative influence that rubs off on the effective leader and renders them ineffective. This situation had been highlighted by 15.4% of the respondents as one of the causes of stagnated DT endeavors in the port's ecosystem, as any newly appointed leaders still had to work with the majority of the mediocre workforce at lower levels. A quality assurance manager had commented that:

> [...] You may find in these governmental parastatals, the demand for skilled labor force is 100 people, but the government allots on 3 or 5 personnel, as such, these personnel however effective they may be, become absorbed by many who are already complacent. ... only introducing a few fresh blood personnel in a system that benefits majority, they invariably become sucked in and go native.

> For instance, we [the quality assurance bureau] faced resistance in digitizing our systems, however due to strength in numbers of new recruits, we were able to overcome resistance of the workforce we found already deeply entrenched in the organization and successfully digitized our systems.
>
> (CM–QREG)

Furthermore, decisions on investments in DT require a level of autonomy Port Omega must exercise. A total of 30.8% of the respondents revealed that a hierarchical leadership style where the port's top executives receive directives from the central government creates an implied fear of reproach. This is argued to have been perpetuating their inability to exercise discretion to undertake investments in DT. Consequently, the port lacks solid IT systems to cater to an ever-growing demand for efficient operations, as a senior official asserted:

> [...] The problem with decision-making in large governmental parastatals like Port Omega is that the guy at the top [port's director general] who is president's appointee, has the highest job insecurity ...his subordinates, both managers and employees, the best they do is shunning being liable too, so, avoiding liability dampens innovation, because very few will dare tell the boss that ...here we have messed up, we need to change the course of action. Instead, all they think about is how they can survive the minister, the secretary general, or the president. Consequently, they hardly have time to seriously plan out the future of the port.
>
> (SO–FFA)

*4.6. Monitoring and Controlling*

This theme related to those actions that the principal executed on the agencies, whose outcomes either promoted or hindered the successful implementation of DT in the port's ecosystem. For instance, the monitoring landscape of Port Omega had been such that it hinged on national political leadership change, political directives, and tenuous leadership style with scarcely specific time-bound deliverables that port leaders must achieve. Table 3 illustrates that 76.9% of all respondents, 10% of whom were internal stakeholders, while 90% were external stakeholders, contributed to the articulation of this theme.

Our analysis indicated that the interference of governmental officials in the management of port operations stifled Port Omega's autonomy. A total of 38.5% of the respondents argued that such interference perpetrated unqualified personnel in critical ports' managerial roles despite lacking an appropriate vision to propel the port digitally. The imposition of top leaders as director general has been argued to cascade fear among the workforce at lower levels of operations, something that has suppressed openness and autonomy in decision-making. A senior IT manager alluded that:

> [...] There is a lot of fear going on at the port, as we speak, ...the top officials are imposed on the port by the government and impart intense anxiety on the workforce. The government thinks by implanting its intelligent security personnel in the port, it will preclude secretive misappropriation of revenues. These people have no idea how the port business works but thrive as the government's whistleblowers, they can hardly make the port progress because they don't possess that vision nor is it a requirement for them to assume such roles. I know of very intelligent colleagues whom if were allowed to make changes to the port, they could have positively moved it further forward.
>
> (SM1–PA)

Similarly, this problem was also highlighted by a customs authority manager who opinionated that the involvement of the government's intelligent personnel in leadership positions in the parastatals impacts strategic decisions necessary to diffuse DT in the port ecosystem. They alluded that:

> [...] Another challenge is that which relates to government's recruitment process for parastatals' leadership. This procedure brings problems in the implementation of governmental projects. For instance, in most top positions the government recruits intelligent security personnel. Now, employ someone as such does not necessarily mean they are competent ... this is where we mess up.
>
> (CM–CA)

Moreover, the results demonstrated that the implementation of digital technologies requires a strategic vision independent of undue political influence from the central government. The respondents argued that the port's top leader's office tenure has been commensurate with the head of state's tenure as they are appointees. They further asserted that the port's directors' tenure hinges on the mercy of the country's president, as affirmed by a senior official:

> [...] Port Omega's top executives are individuals with the highest job insecurities ... they spend most of their time dancing according to the tune of the national leadership in power, trying to appease them, meanwhile, they give away their creativity and ability to steer the port in the right direction.
>
> (SO–FFA)

Our findings revealed that implementing DT in the port ecosystem would enhance monitoring and controlling when agencies' actions can leave an audit trail in the system. For instance, the implementation of eSWS or EDI technologies was argued to optimize Port Omega's operations and provide the government with real-time information on its operations. Accordingly, 53.8% of the respondents argued that this would overcome bureaucratic procedures that bog down port ecosystem efficacy. A senior official alluded that:

> [...] You know, implementing DT would eliminate the bureaucracy that has been causing unnecessary delays and increasing Port Omega's operational costs.
>
> (SO–SAA)

### 4.7. Causal Mechanisms of the Identified Themes

In this section, we provide inferential causal mechanisms of the emerging themes, as discerned by the authors and depicted in Figure 4. Our empirical evidence revealed that, depending on the monitoring approach deployed by the principal to the agencies, adverse selection either attenuates or bolsters the monitoring efficacy. For instance, the appointment of port-related and industry-seasoned top leaders to run the agencies would likely support DT initiatives and their diffusion in the port ecosystem. This would, in turn, have a positive impact on the monitoring outcome. However, the inept port leaders, as described by the respondents, inhibit effective digital strategies that could foster monitoring and controlling mechanisms.

Similarly, adverse selection will likely exacerbate risk aversion and moral hazard. The plausible explanation for this is that inept agency leaders and port ecosystem workforce may shun being responsible by deliberately hiding behind existing disintegrated legacy systems, which may also increase the chances of opportunistic expropriation.

Meanwhile, both risk aversion and moral hazard reinforce each other and exert a greater negative impact on the monitoring efficacy. For instance, the port workforce's fears of reproach and being held accountable likely increase the inclination to sabotage digital systems, heightening moral hazard and risking aversion while waning the monitoring efficacy that would be engendered by such systems. Furthermore, the interplay of moral hazard and risk aversion creates a system where the port workforce is incentivized to maintain the status quo, avoid accountability, and protect their own interests. As revealed by our empirical observation, this has resulted in a reluctance to adopt more efficient monitoring mechanisms, such as implementing digital systems or improving oversight processes, because these changes might have threatened existing power structures or exposed misconduct.

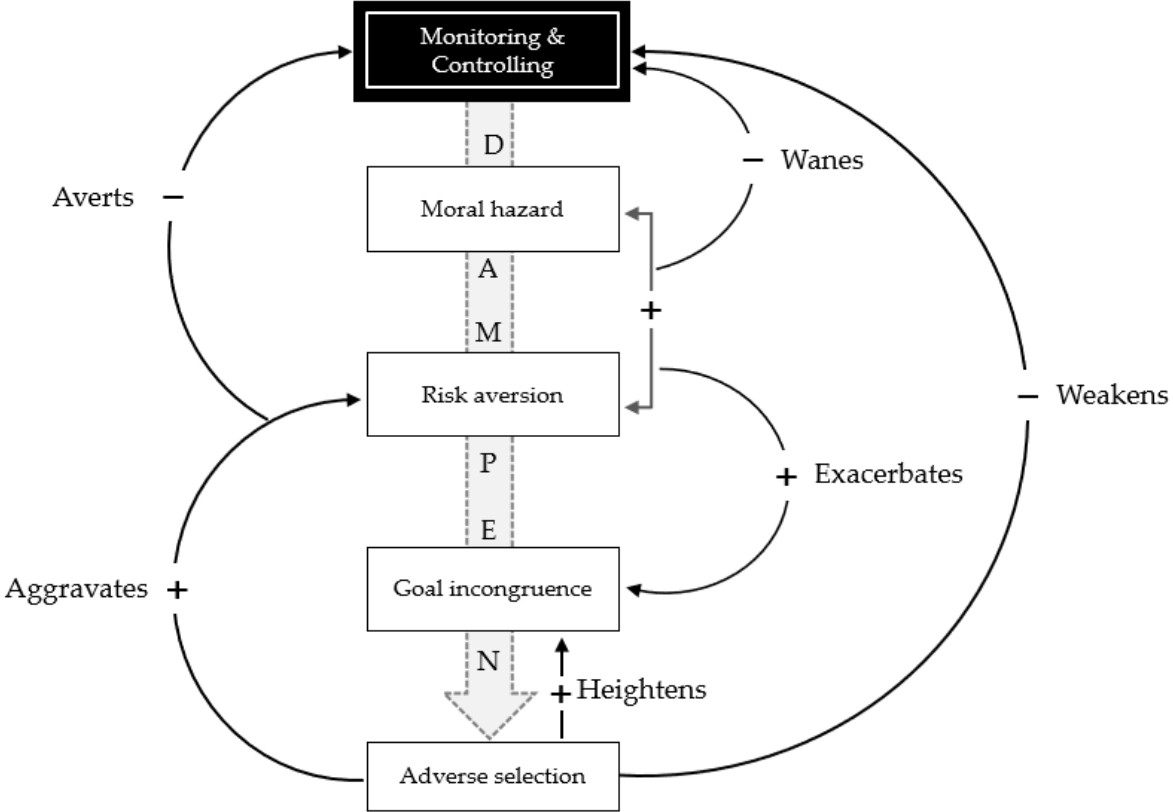

**Figure 4.** Causal mechanisms of the interplay of antecedents challenging successful DT in port ecosystems.

Moreover, the interplay between moral hazard and risk aversion exacerbates goal incongruency among port stakeholders. For instance, as different port departments and other agencies have different cultural orientations, they invariably maintain disintegrated systems. The cultural disconnect and system multiplicities defeat DT penetration and its efficacy in fostering greater monitoring and controlling. Moreover, the interplay of these factors consequently challenges the successful implementation of DT initiatives in port ecosystems. Yet, complacency and a dependence on a few experts result in a lack of innovation and reluctance to adopt new technologies or processes, further exacerbating goal incongruence.

## 5. Discussion

In this section, we place the findings in context and provide both practical and policy recommendations. We test the five propositions against empirical findings presented in Section 4.

This study responds to Jovic et al.'s [20] and Tijan et al.'s [9] calls for empirical work on DT in the maritime industry and nuanced attention to its constitutive sectors—shipping, shipbuilding, and ports—by Raza et al. [3]. Consequently, we focused on ports to gain a richer knowledge of DT and complement recent studies by Raza et al. [3] and Nguyen et al. [35], which investigated DT in the maritime shipping sector. Our findings are based on the investigation of a port in an emerging economy, which augments existing empirical evidence on DT in maritime ports. Our findings demonstrate that the respondents generally perceived eSWS as a DT initiative meant to integrate the port's stakeholders. This understanding affirms the extant literature's depiction of ports' digital maturity trajectories, where basic digital systems form the basis for advanced ones as ports transition into higher levels of DT [3,5]. In response to this study's research questions, our analysis unraveled seven contextual factors implicated in P1–P5 that influence DT endeavors in emerging economies' ports.

Proposition 1 is grounded in the argument from Miller as well as Moore and Vining [47,52] that goal incongruence between principals and agents creates suboptimal outcomes. Our results revealed that a myriad of public authorities with siloed systems have been associated with inter-agency constraints limiting DT penetration. Agencies' disparate objectives that do not converge create conflicts of interest [49]. This exacerbates a lack of definitive digital orchestrators as well as unclear responsibility and cost-benefit sharing accruing from collaborative DT initiatives. The national government requires the port authority and the LPCOs to integrate into an eSWS so as to rationalize documentation formalities and procedures and increase process visibility and monitoring [3]. However, the agencies must contend with the most pressing issues in order of their preferences, which conflict with the national government's objective. As the digital solution is costly and extensive, the ability and willingness of these stakeholders become imperative for effectuating a meaningful DT [69]. This is because these actors are highly intertwined and require an alignment of strategies and cooperation [36]. It follows that goal incongruence challenge requires more concerted efforts that cannot be achieved when each stakeholder pursues their own objective at the expense of the national government. As Carlan et al. [15] argue, the majority of barriers to digital innovation in the port sector could be overcome by embracing the synergistic effects of stakeholders' collaboration.

Furthermore, the system ownership tussle between the customs authority and the port authority exemplifies goal incongruence among dominant stakeholders in the maritime industry. This underscores the complexities of network-related ecosystems experience when it comes to aligning the interests of all stakeholders in an equitable manner. In fact, it is almost impossible to ensure all stakes are not compromised through DT, and therefore, actors must be willing to improvise for the common good. Our findings closely mirror Zeng et al. [24]'s findings that freight forwarders within the shipping sector had been reluctant to adopt an open digital platform for container bookings because of the fear of compromising their discretionary price offers and exposing their trade secrets. Undoubtedly, the use of a common platform as eSWS in port ecosystems eliminates discretionary control that individual actors possess with their own systems. It also introduces a layer of coordination complexity and the problem of multiple principals (i.e., the national government and the actor who will host the system and, therefore, assume power and dictate the terms of access) [32,47]. Consequently, the agent may have to contend with uncoordinated and often conflicting demands, requirements, and incentives [48,70]. Thus, Proposition 1, which states that goal incongruence limits DT progress, is supported.

Proposition 2 deals with moral hazard, a situation where the port authority extracts private rent at the detriment of the national government because the latter cannot directly control the agencies' actions [51]. Our findings revealed that information system projects at the case port have consistently failed due to such acts as system sabotage and corruption by the agency's officials. For instance, the government, as the principal, is ultimately responsible for the funds allocated for digital infrastructure development but may not be aware of or able to prevent the misappropriation of these funds. The national government succumbs to information asymmetries and agencies' commitment problems [51]. Our results indicated that the prevalence of corruption at the port leads to a lack of transparent and integrated systems. The consequence is manual exchanges of documents and inefficient processes, which further aggravate corruption's vicious circle as kickbacks and bribes become necessary catalysts for spurring agencies' officials into action. Likewise, the government may insist on the adoption of digital technologies by port authorities to remedy the problem of moral hazard [71], only to be met with deliberate acts of sabotage and expropriation of project funds. Undoubtedly, the implementation of DT would foster monitoring as processes become visible and traceable. However, greater efforts need to be made to make digital change take root in an environment where manual operations prevail and, for such reason, have become a source of private rent for officials and social inefficiency for the government [47].

Our findings echo Gekara and Nguyen's [25], who revealed failed digital innovation as a result of deliberate system sabotage by Mombasa Port's officials. While the authors

did not allude to corruption, moral hazard is typical in developing economies with weak governance systems [72]. Due to limited transparency in most developing countries, large investments in DT arguably attract embezzlement and misappropriation. Our findings revealed how DT endeavors at the case port stalled over almost a decade due to corruption, sometimes with top leaders involved in unscrupulous contractual agreements with incompetent digital system vendors. This erosion of institutional checks and balances stifles ports' DT trajectories in developing economies. These findings contrast with Gausdal et al.'s [4] work, which explicitly revealed how corruption is frowned upon in the Norwegian maritime industry. The authors further acknowledge that while corruption may not be an issue in developed countries, it is definitely a reality in some countries and may pose a barrier to DT. Meanwhile, the literature on corruption abatement [73] argues that moral hazard will be abated by tying the decentralization of agencies to improvements in income inequality among the workforce. Thus, Proposition 2, which states that agencies' inclination to private rent-seeking stifles investments in DT, is supported.

Proposition 3 deals with the adverse selection between the government and port authority, which results in suboptimal port operations such as cargo handling and logistics [52]. Our results indicated that the port authority relies heavily on manual interventions, which perpetrate information asymmetry. The port authority is known to embrace manual interventions and is slow in transforming its operations digitally. Undoubtedly, manual interventions obscure visibility into port processes [3]. The lack of transparency and manually recorded data may perpetuate a situation where inaccurate or incomplete information is being provided. This can lead to problems when the government relies on this information for decision-making. As the digital system fosters transparency of port operations, ports' internal stakeholders may deliberately shun their implementation in order to continue safeguarding their interests—private rent-seeking. This revelation closely echoes Raza et al. [3], who discovered that the workforce in the liner shipping segment has a longer tenure in office, which results in their inflexibility in adopting DT. The latter requires a digitally adept and vibrant younger workforce.

The impact of a few digitally adept workforce on ports' DT endeavors is questionable because of the possibility of becoming native, as a majority of the workforce may still embrace manual processes, and thus, penetrating the digital agenda may prove futile. For instance, Zeng et al. [24] revealed that humans still performed the bulk of work in the container shipping chain, with freight forwarders recording up to 90% of labor costs. This has complicated the absorption of container booking digital platforms among actors because humans were thought to be better suited to handling issues urgently than digital systems, given the complicated nature of the shipping process. Thus, ports' holistic considerations are a necessity in transforming their workforce's digital attitudes [21], which will then pioneer subsequent DT endeavors. Consequently, Proposition 3 is supported.

Proposition 4 concerns risk aversion when the port authority shuns responsibility for failure relating to negative port outcomes that contradict the central government's goals. Our results indicate that the fear of negative consequences, both individually and organizationally, can lead to a reluctance to embrace change and take calculated risks. People often resist change if they perceive it risky for their personal interest or job security. The proxies for risk aversion have been observed in the port's complacency, fear of reproach, hierarchical decision-making, and cultural influence on leadership outcomes. Our observations complement Raza et al.'s [3] findings that organizational culture restrained DT efforts among liner shipping companies. Together, these findings demonstrate the complexity of the socio-aspects present beyond the mere adoption and implementation of DT. In fact, the cultural constraints aggravate other challenges, such as a decision to invest funds in DT initiatives [4,9,53], and may also be influenced by factors such as goal incongruence and adverse selection [32,48,51]. Meanwhile, the lack of delegation and empowerment at ports' lower levels can also lead to resistance to change. Frontline workers, who often possess valuable insights and hands-on experience, may resent embracing DT

initiatives if they are not actively involved in the decision-making process. This can result in a reluctance to learn and adapt to new technologies, consequently impeding DT progress as employees and managers may prioritize self-preservation over innovative decision-making. Our finding echoes Chowdhury et al.'s [29], who found that the aversion to change presented a significant barrier to the implementation of smart port practices at Chattogram Port Authority. Thus, Proposition 4 on risk aversion is supported.

Proposition 5 argued that the government, as a principle, requires DT in order to bolster its monitoring efforts against information asymmetry and associated moral hazard and adverse selection. While this is a desired objective, as suggested by the PAT literature [32,47,51], our results revealed that excessive control over the port authority seemed to stifle the autonomy and discretion it takes to experiment with novel and evolving DT. We discuss this proposition in light of the national government's political influence on port governance. We found that political interference in the running of the port, where there was no clear demarcation between autonomy and responsibility, defeated the clear DT strategy on the part of the agent. For instance, the fact that port leaders are not recruited on a meritocratic basis but appointed by the national president removes competitiveness in the leadership role. As revealed by empirical data, such leaders lap up the interest of the president in power. The imposed port's top leaders likely instill fear and anxiety among the lower-level workforce, which stifles openness and autonomy in decision-making and exacerbates inclinations to self-serving behaviors among the officials. This observation illustrates how coercive political interference can break DT initiatives in contrast to empirical findings by Kuo et al. [74], who established that coercive pressure significantly affected DT. It also underscores the flexibility with which agents must experiment with DT in the absence of the principal's excessive monitoring and controlling. Likewise, Kashav et al. [57] provide evidence from the Asiatic context that illustrates how political uncertainties and the involvement of politically motivated strategists and policy makers stunt the growth of maritime supply chains. The authors further contend that ports in Europe and other developed economies have successfully circumvented these nuisances and are pioneers in digitalizing their ecosystems.

Furthermore, political influence can serve as an imminent force that propels DT initiatives across a range of port stakeholders, especially those who are state-owned. This is in accordance with the agency theory, where monitoring is expected to ensure an agent lives up to their principal's expectations [47,54]. However, excessive political involvement in the running of the port may impose strategic moves that are prematurely conceived and thus irrelevant to the overall efficacy of the port's ecosystem. While the central government must have an overview of what the port is doing, including an overarching understanding of the port's digitalization efforts, the ultimate autonomy for driving these changes should rest with the top leadership of the port. The latter should define strategic moves that not only incorporate DT into current port operations but also anticipate its role in shaping the port's trajectory for years to come. Strategic objectives must be flexible enough to guide investment decisions in novel technologies and should also survive national leadership transitions [52]. This independence will ensure that hiccups in staying focused are managed with reference to existing strategic directives. Similarly, it will ensure that ports remain agile and forward-looking in their DT's embrace, aligning with government directives while leveraging digital innovation for sustainable growth.

Nonetheless, the decision on the type of digital technology suitable for addressing agencies' needs should rest with the agents themselves. However, as agents assume significant but transient top executive roles, they risk being laid off when failure materializes [47,51], limiting their capacity to exploit DT's affordances. Unfortunately, DT initiatives are not guaranteed to take root [35] and are contingent upon the port's specific context, the complexity of changes involved, and the port's readiness [23]. Therefore, by appointing and sacking top leaders, efforts to diffuse DT through fail-forward iterations [3] may stunt. Arguably, the novelty of DT and the uncertain nature of its implementation require minimum monitoring and controlling in order to allow for discretionary experimentations by

the ports, who are agents in favor of government funds. As Fayezi et al. [54] suggested, digital technologies such as ERP, vendor-managed inventory (VMI), and efficient consumer response (ECR) require a high level of authority delegation for their successful implementation. Consequently, Proposition 5 is supported on the grounds that the monitoring practices at play provide contingent mechanisms that either foster or inhibit DT rooting in port ecosystems. Table 4 provides a summary of our discussions in light of the findings.

**Table 4.** Theoretical integration into empirical evidence.

| | | | Representative Backdrop Empirical Evidence | | | | | | | | | | | | | |
|---|---|---|---|---|---|---|---|---|---|---|---|---|---|---|---|---|
| | | | SM1-PA | OM-PA | SM2-PA | OM-STV | CM-CA | BO-CAG | SO-FFA | SO-SAA | OM-PAREG | CM-QREG | OM-TRD1 | LM-TRD2 | SO-SHC | |
| **Prepositions** | **Theme** | | *Internal* | | | | *External* | | | | | | | | | **Status** |
| P1: | Successful implementation of DT initiatives in seaports is contingent upon the effective alignment of incongruent interests between the national governments and their port-related agencies. | Goal incongruence | | | ▓ | | ▓ | | ▓ | ▓ | ▓ | | | | | Supported |
| P2: | Inclination to misappropriate illicit rents by port authorities constrains ports' DT endeavors due to perceived transparency and accountability digital systems enforce. | Moral hazard | ▓ | | | | | | ▓ | ▓ | | | ▓ | | ▓ | Supported |
| P3: | Port authorities whose top leaders are not keen to streamline their processes demonstrate a limited DT footprint. | Adverse selection | ▓ | | | ▓ | ▓ | | | | | | | ▓ | | Supported |
| P4: | Different risk perceptions between national governments and port-related agencies attenuate disinclination to the risk appetite it takes to promote DT footprint in seaports. | Risk aversion | | | | | | ▓ | | ▓ | | ▓ | | ▓ | | Supported |
| P5: | Successful implementation of DT in ports will enhance principals' monitoring efficacy in reducing agency costs; however, this relationship is contingent upon the nature of principals' monitoring practices. | Monitoring and controlling | ▓ | | | | ▓ | | ▓ | ▓ | | | | | | Supported |

The coloured cells indicate corresponding empirical evidence supporting P1–P5.

## 6. Conclusions

Important lessons can be drawn from this study. Firstly, as the five propositions have been informed by empirical evidence, theoretical insights, and practical considerations, this study offers valuable perspectives for both practitioners and scholars in the realm of port management and maritime economics. Secondly, the rigorous analysis of empirical data and critical reflections on the nature of agency dynamics in a maritime port contribute to a deeper understanding of the challenges inherent in implementing DT successfully. Thirdly, ports in emerging economies may likely be at their nascent stage of implementing DT. Typically, these ports may be striving to transition into paperless states with basic institutions of digital platforms such as eSWS, as has been the case with Port Omega. To successfully navigate through higher levels of DT maturity, these ports must develop an affirmative frame of reference (i.e., digital strategy) to guide through implementations of digital solutions. By considering the challenges we have highlighted in this study, port practitioners and stakeholders may take necessary actions to safeguard accelerated DT endeavors in their ecosystems. Thus, a more pragmatic approach to integrating ports' digital resources with external stakeholders is warranted. Consequently, the researchers offer a framework that will guide the implementation of digital solutions in ports in emerging economies by minimizing the challenges we have empirically highlighted. The framework recommends four stages that port leaders and associated stakeholders should undertake to effectively undertake DT. These stages include the following: (1) port ecosystems value chain mapping, (2) stakeholder engagement, (3) resource mobilization, and (4) effective monitoring, as illustrated in Figure 5. Our framework addresses the issue that He et al. [37] raised regarding limited governance strategy, which coheres with DT initiatives.

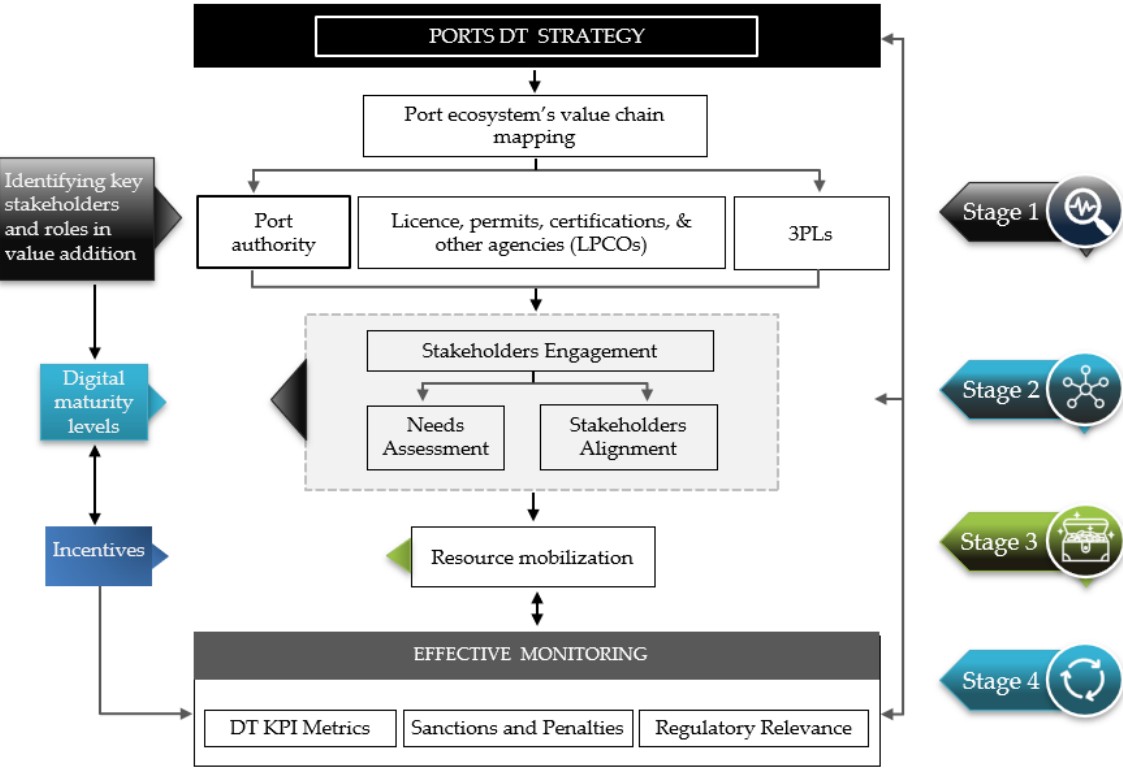

**Figure 5.** A holistic framework of DT strategy in port ecosystems in emerging economies.

### 6.1. Port Ecosystem's Value Chain Mapping

The first stage of establishing the port's digital strategy should begin with a holistic approach to mapping all relevant stakeholders with a bearing on port operations and value addition beyond the confines of respective ports. During this process, such aspects as digital orchestrators and innovation champions must be established. Stakeholders must

then be categorically grouped according to their role in value addition at ports. Moreover, the process should be inclusive in order to entice the buy-in of relevant stakeholders who may otherwise opt out due to their limited involvements in port-related issues. The latter, as suggested in the extant literature, is necessary to invoke responsible practices on stakeholders. Thus, stakeholders such as customs authorities, shipping lines, clearing and forwarding agents, terminal operators, container freight stations, inland container depots, and all other agencies mandated to ensure quality conformance of imports and exports, commonly known as LPCOs, must be given the utmost consideration. In the mapping process, all communication bottlenecks should also be identified.

### 6.2. Stakeholders Engagement

After mapping the port's value chain, the next stage is to solicit stakeholder engagement. This is critical as port efficacies are also defined by the extent to which pertinent stakeholders are effective, and the concept of the weakest link in the chain cannot be overemphasized [43]. As the extant literature has it, the interdependencies among stakeholders complicate the process of DT to the extent of divergent interests [32] and the desire to curb the infiltration of trade secrets [24]. Here, two groups of stakeholders must be considered: public stakeholders (i.e., governmental agencies) and private stakeholders (i.e., terminal operators, three PLs, clearing and forwarding agents, and shipping lines). The actual roles of each stakeholder in DT initiatives must be significant. To ensure this, two critical activities must be undertaken. First is each stakeholder's need assessment, and second is the alignment efforts that are required to onboard them on the holistic port DT strategy. In the process of need assessment, activities that each stakeholder group performs must be identified, and their digitization extent must be ascertained. DT maturity levels should be given priority as stakeholders may have many other pressing needs. Moreover, stakeholders' cultural values, societal interests, and concerns must be taken into consideration [75]. Once this has been achieved, the costs and benefits that may accrue from such engagements must be estimated and explicitly communicated to all involved stakeholders. Consequently, either a collaborative governance framework or a neutral governance organization [35] must be developed. This will help mitigate interagency constraints and ownership tussles, as all relevant stakeholders must have been in the decision-making processes related to DT initiatives. Thus, stimulating collaboration and cooperation of port stakeholders in digital initiatives are a necessary condition for the success of DT strategy and, therefore, its diffusion [4,36,41]. Similarly, the significance of stakeholders' roles in DT initiatives will largely determine its success [35] and overcome such challenges as system sabotage and siloed systems.

### 6.3. Resource Mobilization

During the process of resource mobilization, port stakeholders move to the most critical phase that abounds the extant literature as an imminent stumbling block to DT diffusion in port ecosystems. To dampen the impact of investment cost on DT outcomes, port ecosystems must implement relatively matching investment priorities commensurate with the stake they have in port value addition. For instance, either customs authorities or port authorities may take on a DT orchestrator role in spearheading digital initiatives [28]. Once this has been achieved, stakeholders must align their digitalization efforts. In the meantime, depending on the ownership structure of port ecosystems, incentive schemes must be implemented on an outcomes basis. For instance, stakeholders who can evidence implementation of DT initiatives should be offered certain monetary incentives such as tax reliefs. Grants may also instigate greater DT diffusion in port ecosystems; however, these should be equally accessible to all stakeholders, and should not come up with too-stringent measures in case they defeat digital experimentation. This will foster the participation of pertinent stakeholders in the DT process who otherwise differ in terms of the benefits and resources at their disposal [28].

While port ecosystems cannot achieve DT at once, capacity-building and skill development programs must be established to equip the port ecosystem's workforce with the knowledge and expertise to effectively leverage digital solutions [41]. This can empower employees to embrace DT. Moreover, linkages and partnerships with technology providers who are the critical capability providers [15], and academia become imperative avenues from where port ecosystems can access expertise, digital resources, and innovative solutions, thus accelerating the diffusion of DT and minimizing the problem of talent poaching [4] due to limited digitally adept workforce. Port authorities or customs authorities, whoever is the DT's orchestrator, must proactively entice and involve other stakeholders in the bid to win their interest and dedication [28].

### 6.4. Effective Monitoring

Finally, the port's DT strategy will be effective to the extent of robust monitoring schemes. And here, the national governments, through their respective ministries, such as the Ministry of Information and Telecommunication Technology and the Ministry of Transport Infrastructure, have key roles to play. DT of port ecosystems is not the sole responsibility of ports only, as in resource mobilization, an independent regulatory body should be charged with the responsibility of ensuring the port ecosystem goes digital one step at a time. This regulatory body should, together with all pertinent stakeholders, establish DT KPIs metrics, which will enable it to be impartial to any stakeholder when evaluating its DT initiatives. The independent regulatory body must then exercise the power vested in it to sanction any stakeholder who will seem to perform poorly with respect to going digital. It should also provide recommendations on best practices and liaise with the government on matters relating to regulatory relevance. This is particularly important to overcome the issue relating to information sharing and prevalent disintegrated systems among stakeholders, particularly the LPCOs. As indicated in the extant literature, regulatory support has a significantly positive impact on innovation adoption [38].

Moreover, each stakeholder group should have its own digital champions and whistle-blowers who keep an eye out on all other stakeholders and provide a feedback loop on digitalization challenges they still experience as they interact with the port on a daily basis. This will ensure that stakeholders snap out of complacency and embrace a more transformative culture. The saying that it takes two to tangle cannot be overemphasized here as far as corruption goes. In this regard, sensitization programs must be instituted, and whistleblowing should be effective so that individuals or organizations involved in corruption scandals do not get away with it. Moreover, sensitization programs may overcome the tendency of stakeholders to sabotage systems in the quest to smother transparency and accountability. As Philipp [41] argues, digital awareness allows sensing and strategic investment in digital innovations, and sensitization programs become indispensable. Similarly, stakeholders, both governments and agencies, must establish and adhere to the code of conduct and standard operating procedures, which must have been established collaboratively in the stakeholder engagement stage. This will help mitigate excessive political influence on agencies' governance outcomes, ensuring that decisions on DT are made meritocratically.

It is important to stress that the recommendations in this framework may only become practical if followed through persistently for several years. It may take a while to gain alignment with stakeholders whose interests may be compromised by the initiatives we have recommended herein.

### 6.5. Theoretical Implication

From a theoretical perspective, this study is one of the early efforts to demonstrate the applicability of PAT in the maritime industry's DT. DT can bolster the monitoring and controlling exercised by principals to dampen the effects of moral hazard, risk aversion, adverse selection, and goal incongruence of agents [32]. However, the relationship between DT and monitoring efficacy is not determinate as it is contingent upon already existing monitoring practices; in reality, we have demonstrated that excessive monitoring of agents

by the principal eliminates the former's autonomy and discretion on matters relating to DT. Furthermore, stakeholders' differing information availability [32] may likely contradict DT endeavors in port ecosystems.

### 6.6. Limitations and Future Research Directions

This research has provided invaluably in-depth insights into DT's challenges in a maritime port in a developing country. This value-laden approach is particularly problematic for positivistic purists who are all out for the objective generation of universal theories [65] from an inquiry. This problem, however, is valid only at face value as far as the sacrosanct views of the positivists are at play. Otherwise, this study has provided context-dependent empirical evidence appropriate for unpacking the DT phenomenon in its nascence. In fact, constructivists argue that qualitative studies are valuable for what they are and provide their own unique strengths that are not necessarily pejorative when juxtaposed against hypothetico-deductive approaches. Thus, Flyvbjerg [76] argues that formal generalization alone does not qualify the scientific rigor of an inquiry but the depth and breadth of analysis and transparency of the entire research process. Throughout the manuscript, this study has embraced the latter, and with the transparent research process, the study can be reproduced in similar contexts or, better yet, used as a benchmark for comparative analyses thereof.

Moreover, our integration of qualitative empirical evidence into a well-established universal PAT helps eradicate one of the common misconceptions about case studies, as argued by Flyvbjerg [76], that they can hardly be used to test existing theories. Importantly, concerns about the generalizability of this study are mitigated considering our strategic decision to use the port ecosystem's value chain actors. This means that drawing conclusions on the narratives about these value chain actors serves all practical purposes of understanding stakeholders' actions, meanings, and processes as the DT phenomenon unfolds in the ecosystem. Therefore, the use of multiple respondents and secondary sources of data in triangulating and corroborating our analysis validates our findings and solidifies our conclusions. While the insights derived are context-specific, it is safe to assume that they exert a certain level of external validity or, rather, transferability, as commonly referred to in qualitative research [76], to ports existing in emerging economies contexts. However, this cursory presumption must be guided and supported with further empirical evidence, bearing in mind that generalizability in social inquiries is hardly ever certain [76]. Consequently, this study offers several avenues for further research endeavors. Firstly, future studies may investigate how ports' ecosystems in emerging economies can sustain successful DT initiatives. Secondly, a comparative analysis of ports in other developing economies or across different regions will strengthen the generalizability of the findings if the latter is the only purpose to be achieved [65]. Thirdly, the deployment of mixed method design, which combines qualitative and quantitative approaches, could further provide more robust results regarding DT trajectories in the ports sector. Fourthly, structural equation modeling techniques could ensue from the causal mechanisms of the challenges of DT we have uncovered, thus contributing to advancing knowledge and enriching scientific inquiry.

Unfortunately, the researchers faced the dilemma between maintaining the confidentiality of the case study's respondents and demonstrating transparency in the research process. With respect to confidentiality measures, the authors anonymized sensitive information such as names, locations, and specific details about individuals and organizations involved in the case study. While these actions have been necessary to protect the privacy and integrity of the respondents and fulfill ethical considerations, they inadvertently come across as obscurant. However, to dampen this ineluctable drawback, we have ensured that we exercised transparent reporting and provided sufficient contextual details about the case study.

**Author Contributions:** Conceptualization, B.M.S. and B.I.H.; methodology, B.M.S.; software, B.M.S.; validation, B.M.S., B.I.H. and S.B.; formal analysis, B.M.S.; investigation, B.M.S.; resources, B.I.H. and

S.B.; data curation, B.M.S.; writing—original draft preparation, B.M.S.; writing—review and editing, B.M.S., B.I.H. and S.B.; visualization, B.M.S.; supervision, B.I.H. and S.B.; project administration, B.M.S. and B.I.H. All authors have read and agreed to the published version of the manuscript.

**Funding:** This research received no external funding.

**Institutional Review Board Statement:** Not applicable.

**Informed Consent Statement:** Not applicable.

**Data Availability Statement:** The authors will make the dataset available upon duly request.

**Conflicts of Interest:** The authors declare no conflicts of interest.

## Appendix A. Case Study Protocol

Background information

The extant literature purports that digital transformation (DT) of maritime ports is slow and unforthcoming. Still, most research provides anecdotal evidence on the slowness of ports in implementing DT initiatives. However, there is little empirical evidence on the subject matter that validates such anecdotal evidence. Consequently, the current body of knowledge lacks empirical rigor (Tijan et al., 2021). We particularly center our argument on the fact that, despite the increasing importance of DT in the maritime industry, the lack of comprehensive studies that provide insights into the practical implementation and impact of DT on the various stakeholders in maritime ports exacerbates the understanding of its nuances and granularities. Therefore, the primary objective of this case study is to address this existing gap.

Case study design

In connection with the objective above, this study adopts an embedded single-case study approach. The latter is chosen to allow for an in-depth exploration of the DT phenomenon within Port Omega's natural setting and to understand the complexities of the DT process from stakeholders' vantage points. This design will facilitate the identification of critical factors influencing DT and their impact on the overall performance of maritime ports. As a matter of fact, the design allows for the assimilation of data from multiple sources, such as interviews, observations, and archival records. We use Port Omega's internal and external stakeholders to uncover its DT journey through not only semi-structured interviews but also through document analysis, observation, and official website reviews.

Case selection criteria

As stated in the preceding section, this study uses insights from Port Omega's internal workforce and its key external stakeholders to explore the DT phenomenon. These stakeholders play pivotal roles in Port Omega's DT journey. Therefore, the cases are selected on the strength of information richness and knowledge of port processes that they have acquired through mutual interactions with it. The use of external port value chain actors is designed to ensure the credibility of our findings. Furthermore, external actors, such as clearing and forwarding agents, shipping agents, and logistics service providers, provide an objective and unbiased perspective of Port Omega's operations and DT efforts than internal stakeholders who may have vested interests and so provide lopsided insights. Importantly, external value chain actors help validate Port Omega's responses on its DT journey, which may be misrepresented and biased. By cross-validating Port Omega's responses through the lens of its stakeholders, we will gain a more accurate and comprehensive understanding of the DT phenomenon in this context.

Data collection

The primary source of data for this study will be semi-structured interviews. This approach allows us to accommodate new information that emerges as the interview process unfolds. However, we triangulate information garnered through interviews with Port Omega's publicly available information, official website reviews, and port visits to gain immersion.

Interview guide

Implementation of Digital Transformation at Port Omega: Views from its internal and external stakeholders

Dear Participants,

This interview guide is about how Port Omega and its stakeholders embrace technological transformations in order to streamline interactions and value-creation processes in the ecosystem. Kindly provide your viewpoint about several aspects pertaining to the subject matter in this interview guide. Thank you in advance for taking the time out to read through this interview guide and be part of this research process.

| | |
|---|---|
| 1 | Consider cargo clearing process at Port Omega; in your opinion, what have been the major bottlenecks to efficient clearing process?<br>⇨ How has your organization been reacting to such bottlenecks?<br>⇨ How is your organization attuning itself to overcome such bottlenecks? |
| 2 | How do you think Port Omega, as an organization, has been prepared to implement digital transformation? |
| 3 | What is your opinion about the knowledge of digital transformation Port Omega's executives and staff have?<br>⇨ How digitally adept is the port's workforce and how is the literacy rate impacting digital transformation initiatives? |
| 4 | Port Omega is relatively complex, what do you think have been the greatest challenges so far as interactions with stakeholders are concerned?<br>⇨ To what extent do stakeholders' disparate systems impact Port Omega's efficacy?<br>⇨ Which technological solution(s) do you think will help overcome these challenges?<br>⇨ If the Port Omega adopts such solution(s), what implications will it have on its key stakeholders? |
| 5 | What is the current landscape of integration of other stakeholders in the port's information systems?<br>⇨ How does the port coordinate its operations with other stakeholders?<br>⇨ How efficient is this coordination?<br>⇨ What would you suggest as a plausible solution to inefficiency(ies) if any?<br>⇨ Do you envision a state where the port will achieve full transparency when all stakeholders can share data openly? |
| 6 | How do you believe organizational culture influences the actions of Port Omega's stakeholders' implementation of digital transformation initiatives?<br>⇨ In your opinion, what cultural aspects have had a negative impact on digital transformation initiatives at the port?<br>⇨ How would you suggest these aspects otherwise? |
| 7 | What role do you think the government plays in fostering implementation of digital transformation in Tanzania's maritime industry? |
| 8 | How do you view top executives' support of digital transformation initiatives (if any) at Port Omega?<br>⇨ In your opinion, how do they demonstrate digital leadership orientation? |

Data analysis

A thematic analysis was employed to identify recurring patterns, themes, and connections within the data. We also used abductive reasoning during coding processes. The abductive approach to thematic analysis allows pattern identification in the datasets during analysis while at the same time linking these patterns to well-established theoretical concepts. We used the NVIVO v20.1 software to analyze transcribed data. The use of this software allows us to assimilate and articulate the dataset in a more systematic way, thus increasing the validity and reliability of our results.

Reporting

Findings are reported and supported with narrative verbatim quotes as we travel through the depth of discussion, positioning the study in the existing body of knowledge.

**Appendix B. Frequency Distribution Table of Interview Responses**

| | Inductive Codes | Theme | SM1–PA | OM–PA | SM2–PA | OM–STV | CM–CA | BO–CAG | SO–FFA | SO–SAA | OM–PAREG | CM–QREG | OM–TRD1 | LM–TRD2 | SO–SHC | (f) | % |
|---|---|---|---|---|---|---|---|---|---|---|---|---|---|---|---|---|---|
| | | | Internal | | | | External | | | | | | | | | | |
| 1 | Duplication of documentation requirements. | Goal incongruence | | | | x | x | x | x | | x | | | | | 5 | 38.5 |
| 2 | Misaligned digitalization efforts. | | | | | | x | | | | | x | | | | 2 | 15.4 |
| 3 | Mismatching investment priorities. | | | | | | x | | x | | | x | | | | 3 | 23.1 |
| 4 | Regulatory mismatch of governmental agencies. | | | | | | x | | x | | | x | | | | 3 | 23.1 |
| 5 | System multiplicities/disintegrated systems. | | | x | x | | | | x | x | | | x | x | x | 7 | 53.8 |
| 6 | Independent digital system governing agency. | | | | x | | | | | | x | x | | | | 3 | 23.1 |
| 7 | Infiltration of intra-agency flexibility. | | | | | | | | | x | x | | | | | 2 | 15.4 |
| 8 | Shunning compound delays blame. | | | | | | | | | | x | | | | | 1 | 7.7 |
| 9 | Unclear digital hand-over strategy. | | | | x | | | | | | x | | | | | 2 | 15.4 |
| | | | | | | | | | | | | | | | | **12** | **92.3** |
| 10 | Fear of being laid off. | Moral hazard | | | | x | | | | | | | | | | 1 | 7.7 |
| 11 | Fear of being held accountable. | | x | | | | | | x | | | | | x | | 3 | 23.1 |
| 12 | Fear of relinquishing institutional and personal power. | | | | | | | | | | | | | | x | 1 | 7.7 |
| 13 | Bureaucracy. | | | | | | | x | | x | | | | | x | 3 | 23.1 |
| 14 | Embezzling project funds. | | | | | | x | | | | | | | | | 1 | 7.7 |
| 15 | Incentivising officials to hasten clearing formalities. | | | | | | | | x | | | x | x | | | 3 | 23.1 |
| 16 | Misappropriating IT investment funds. | | | | | | | | | x | | | | | | 1 | 7.7 |
| 17 | Misusing cargo inspection scanners for personal gain. | | | | | | | | x | | | | | | | 1 | 7.7 |
| | | | | | | | | | | | | | | | | **10** | **76.9** |
| 18 | Higher dependence rate on a few IT experts. | Adverse selection | | | | | x | | | | | | | x | | 2 | 15.4 |
| 19 | Manual location process. | | | | x | x | | x | | | x | x | | | | 5 | 38.5 |
| 20 | Manual verification process. | | | | x | x | | x | | x | | x | x | | | 6 | 46.2 |
| 21 | Repeated handling of same documents across disintegrated internal systems. | | | x | x | | | | | | | | | x | x | 4 | 30.8 |

| | Inductive Codes | Theme | SM1–PA | OM–PA | SM2–PA | OM–STV | CM–CA | BO–CAG | SO–FFA | SO–SAA | OM–PAREG | CM–QREG | OM–TRD1 | LM–TRD2 | SO–SHC | (f) | % |
|---|---|---|---|---|---|---|---|---|---|---|---|---|---|---|---|---|---|
| | | | | Internal | | | | | | | External | | | | | | |
| | | | | | | | | | | | | | | | | **11** | **84.6** |
| 22 | Complacency. | | x | x | | | | x | x | x | | | | x | x | 7 | 53.8 |
| 23 | Cultural influence on leadership outcomes. | | x | | | | x | | | | | | | | | 2 | 15.4 |
| 24 | Fear of reproach. | Risk aversion | x | | | | | | x | | | | | | | 2 | 15.4 |
| 25 | Having bad attitude | | | | | | | | | x | | | | | | 1 | 7.1 |
| 26 | Hierarchical culture. | | | x | | | x | | x | | | x | | | | 4 | 30.8 |
| 27 | Intra-departmental communication silos. | | | x | x | | | | | | | | | | | 2 | 15.4 |
| | | | | | | | | | | | | | | | | **10** | **76.9** |
| 28 | Implementing eSWS | | | | | x | x | x | x | x | x | | x | | | 7 | 53.8 |
| 29 | National leadership change. | Monitoring and controlling | x | | | | | | | | | x | | | | 2 | 15.4 |
| 30 | Political directives. | | x | | | | x | | x | | | x | | | x | 5 | 38.5 |
| 31 | Tenuous leadership. | | x | | | | | | x | | | | | | | 2 | 15.4 |

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
