# Peer review of "The Principal-Agent Theoretical Ramifications on Digital Transformation of Ports in Emerging Economies"

_logistics, 2024_

Round 1

Reviewer 1 Report

Comments and Suggestions for Authors

Review Comments

logistics-2866593

Title: The principal-agent theoretical ramifications on digital transformation of ports in emerging economies

Keywords look like very generic terms such as “Case study, Maritime industry, Theory elaboration”.  The authors are suggested to give unique terms in relation to this paper.

In Line No.70, “principal-agent theory (PAT)” -----Use appropriate citation and reference to this term.

In Literature and other sections, most of the sentences start with [Reference Number] and it is appropriate to mention the first author’s name followed by the [Reference Number] if the sentence is supposed to start with the author's name. Otherwise, there is no need for author’s name inside the text elsewhere for citation. This is the usual practice.

In Figure 1. “PAT framework of DT’s challenges in the maritime ports sector”….The letters are not legible inside the figure…Please correct them to be readable.

In line No.219 and there on, the authors have mentioned “Port Omega” that was not properly cited throughout this paper.

In Line No. 263-265, “publicly available information and documents such as port’s handbook, online news articles, and stakeholders’ official websites”………….Where is it available? The source could be finally given in the manuscript.

Section 4. Results look very lengthy. It is always appropriate to shorten the text and give it in tabular or pictorial representation along with inferences.

More over the idea behind the propositions are to be explained initially. In the results it could be shortened with few illustrations.

In Conclusion …………since it is a theoretical and survey study, the authors must state about the implementation of their recommendations in a systematic manner and it is a large logistics handling sector, the outcome is so important that must be clearly explained in the section.

Author Response

Dear Reviewer,

Thank you for your feedback on our manuscript. We appreciate your attention to detail and your valuable insights. In light of your suggestions, we have carefully improved the various aspects of the manuscript.

Faithfully, 

Reviewees.

Reviewer 2 Report

Comments and Suggestions for Authors

This is a very interesting paper on a topic that addresses a knowledge gap Overall, the paper can be accepted with minor revision. A couple of points that would be relevant to address:

- Section 3.1 Case description: It would be relevant to include more information (eventually in an annex) about Port Omega + a couple of tables with key indicators of the activities in the Port over time + perhaps a map showing the location of the Port

- Sections 4 and 5: Further explanation for why Proposition 5 is not supported would be relevant. Proposition 5 states that successful implementation of DT will enhance the principal's monitoring and reduce agency costs and I am not sure that the empirical findings necessarily goes against that proposition. The empirical findings shows that excessive monitoring would not be optimal. However, I would still think that in case DT is successful it would enhance the principal's monitoring while reducing agency costs? 

- The subsection headers in Section 4 should be checked: There are 2 subsections both referred to as 4.3, while there is no section 4.4?

Comments on the Quality of English Language

Overall, quality of English is high. Only a couple of typos to be corrected etc.

Author Response

Dear Reviewer,

Thank you immensely for your comments on our manuscript. We appreciate your  invaluable insights. In light of your suggestions, we have carefully improved the various aspects of the manuscript.

Faithfully, 

Reviewees.

Reviewer 3 Report

Comments and Suggestions for Authors

Thanks for submitting the paper. Dear Authors, I have some comments which are:

1. The abstract is very weak. The author must remember to understand what the Abstract should contain. It may have a little introduction to the field, what is the problem or challenge, your proposed solution for the problem, and the main highlights of your solution. High-level discussion on results, maybe two or so lines.

2. The method needs to be stronger, with self-claims without such supported literature. There is no analysis; where it is, all is theoretical with no or weak evidential support.

3. Results should be represented in a more robust manner. I hardly see any qualitative matter; all are self-claims.

Comments on the Quality of English Language

Can we proofread once.

Author Response

Dear Reviewer,

Thank you for your feedback on our manuscript. We are thankful to you for your invaluable comments and insights. In light of your suggestions, we have carefully improved the various aspects of the manuscript.

Faithfully, 
Reviewees.

Reviewer 4 Report

Comments and Suggestions for Authors

I have reviewed the study entitled “The principal-agent theoretical ramifications on digital transformation of ports in emerging economies”. Although the discussion and conclusion sections of the study were well organized, there are some major issues in the introduction section. Therefore, I recommend a major revision.

1) The introduction section is far from presenting the required theoretical discussions on the topic. Therefore, theoretical arguments need to be strengthened.

2) The primary motivation of the study was not clear.

3) The literature gap was ill-presented.

4) The discussion and conclusion sections were well organized.

5) There are some grammatical errors in the study.

6) The total similarity level of the study was 4%. Be sure that the revised version of the study should have no more similarity level than 20%.

Comments on the Quality of English Language

Minor editing of English language required

Author Response

Dear Reviewer, 

Thank you for your feedback on our manuscript. We appreciate your invaluable contributions and insights. In light of your suggestions, we have carefully improved the various aspects of the manuscript.

Faithfully, 
Reviewees.

Round 2

Reviewer 3 Report

Comments and Suggestions for Authors

Dear Authors, I have some comments:

You improved the work; the abstract still needs to be improved, and the result section needs to be stronger. Authors need to understand that the study is working on assumptions; now, at least, I would expect solid results and an analysis section that can be quantitative and qualitative. However, at least quantitative.

The abstract is the fusions and holistic explanation of the "Doman introduction, Problem, proposed solution, and Results, may be a comparison in Results.

Comments on the Quality of English Language

A final proofread is needed.

Author Response

Dear Reviewer, 

We want to express our sincere gratitude for your thoughtful and insightful review of our manuscript. Your feedback and constructive criticism have been invaluable in improving the quality and clarity of our research. 

Please see the attachment for further reference.

Thank you once again for your invaluable contributions.

Sincerely, 

Reviewees.

Reviewer 4 Report

Comments and Suggestions for Authors

Authors have mostly addressed former queries. Therefore, editorial can consider related paper for potential publication.

Comments on the Quality of English Language

 Minor editing of English language required

Author Response

Dear Reviewer, 

Once again, thank you immensely for your invaluable inputs that have enabled us to significantly improve the manuscript. Thank you also for recognizing efforts exerted in the improved manuscript, and for commending it for publication consideration in Logistics journal. 

Sincerely, 

Reviewees. 

Round 3

Reviewer 3 Report

Comments and Suggestions for Authors

Dear authors, thanks for addressing the comments. I am satisfied with your comments, as you addressed all major concerns. 

Comments on the Quality of English Language

One final proofread needed.